# Isentropic Mixing vs. Convection in CLaMS-3.0/MESSy: Evaluation Using Satellite Climatologies and In Situ Carbon Monoxide Observations

Paul Konopka<sup>1</sup>, Felix Ploeger<sup>1,2</sup>, Francesco D'Amato<sup>4</sup>, Teresa Campos<sup>7</sup>, Marc von Hobe<sup>1</sup>, Shawn B. Honomichl<sup>7</sup>, Peter Hoor<sup>3</sup>, Laura L. Pan<sup>7</sup>, Michelle L. Santee<sup>5</sup>, Silvia Viciani<sup>4</sup>, Kaley A. Walker<sup>6</sup>, and Michaela I. Hegglin<sup>1,2,8</sup>

**Abstract.** Lagrangian modeling of transport, as implemented in the Chemical Lagrangian Model of the Stratosphere (CLaMS), connects the advective (reversible) component of transport along 3D trajectories with mixing, the irreversible component. Here, we investigate the interplay between strongly localized convective uplifts and large-scale flow dynamics in the upper troposphere and lower stratosphere (UTLS). We revisit the Lagrangian formulation of convection in CLaMS-3.0/MESSy, driven by ECMWF's ERA5 reanalysis, and further develop the model. These developments include refining spatial resolution in the Planetary Boundary Layer (PBL) and decoupling the frequency of the adaptive grid procedure—which captures isentropic mixing and redefines Lagrangian air parcels—from the parameterization of convection.

To improve the model's UTLS transport representation, particularly from the PBL over days to weeks, we derive zonally and seasonally resolved climatologies of CO partial columns (XCO, spanning 147 to 68 hPa) and compare them with Microwave Limb Sounder (MLS) and Atmospheric Chemistry Experiment Fourier Transform Spectrometer (ACE-FTS) observations, as well as in situ data. Incorporating a parameterization for unresolved convection significantly improves CO anomaly representation in the UTLS, particularly in capturing seasonal and spatial patterns. While the simulated absolute XCO values align better with ACE-FTS, the model reproduces MLS anomalies more accurately, suggesting MLS better represents CO variability. In situ observations in the boreal polar region generally support lower ACE-FTS CO values, while MLS better represents CO enhancements in air affected by the Asian summer monsoon above 10 km.

#### 1 Introduction

The Chemical Lagrangian Model of the Stratosphere (CLaMS) is a Chemistry Transport Model that employs an irregular Lagrangian grid, trajectory-based 3D advection driven by diabatic, cross-isentropic vertical velocities, and an adaptive grid

<sup>&</sup>lt;sup>1</sup>Institute of Climate and Energy Systems, Stratosphere (ICE-4), Forschungszentrum Jülich, Germany

<sup>&</sup>lt;sup>2</sup>Institute for Atmospheric and Environmental Research, University of Wuppertal, Wuppertal, Germany

<sup>&</sup>lt;sup>3</sup>Institute for Atmospheric Physics, Johannes Gutenberg University, Mainz, Germany

<sup>&</sup>lt;sup>4</sup>National Institute of Optics, CNR-INO, Via Madonna del Piano 10, Sesto Fiorentino, Florence, Italy

<sup>&</sup>lt;sup>5</sup>Jet Propulsion Laboratory, California Institute of Technology, Pasadena, California, USA

<sup>&</sup>lt;sup>6</sup>Department of Physics, University of Toronto, Toronto, Canada

<sup>&</sup>lt;sup>7</sup>National Center for Atmospheric Research, Boulder, Colorado, USA

<sup>&</sup>lt;sup>8</sup>Department of Meteorology, University of Reading, Reading, UK

procedure to simulate isentropic mixing. In this study, CLaMS is driven by ERA5 reanalysis data from ECMWF (Hersbach et al., 2020). Initially formulated as a purely stratospheric, isentropic model on selected potential temperature ( $\theta$ ) levels, early CLaMS applications focused on ozone depletion in the polar vortex (McKenna et al., 2002b, a).

Despite the clear advantages of the Lagrangian approach, which inherently separates mixing from the advective part of transport, very few global Lagrangian chemical transport models with explicit mixing exist (Reithmeier and Sausen, 2000; Stenke et al., 2008; Wohltmann and Rex, 2009; Pugh et al., 2012). One of the key challenges in atmospheric transport modeling is the accurate representation of mixing and convection, particularly in the troposphere and at the interface with the stratosphere. Recent studies have shown that the persistent moist bias in the lowermost stratosphere observed in current climate models (Stenke et al., 2008, 2009), often exceeding 200%, is strongly influenced by the choice of transport schemes. Eulerian transport models tend to introduce excessive numerical diffusion, whereas less diffusive Lagrangian schemes, such as the one implemented in CLaMS, significantly reduce this bias, leading to improved climate simulations (Charlesworth et al., 2023; Ploeger et al., 2024). The latest version, CLaMS-2.0/MESSy, extends the model to include the troposphere and utilizes the Modular Earth Submodel System (MESSy) as its software framework (Jöckel et al., 2010). This version also runs on the JUWELS supercomputer (Jülich Supercomputing Centre, 2019; Konopka et al., 2019, 2022).

The aim of this paper is to revisit and improve the model formulation of unresolved convection, particularly its spatial and temporal resolution, and its interaction with the CLaMS mixing parameterization, which is based on the adaptive grid procedure (Konopka et al., 2004; Pommrich et al., 2014). In the first part of the paper, we review all CLaMS components related to transport processes. In the second part, we evaluate this new model version, CLaMS-3.0/MESSy, using a particularly well-suited diagnostic to assess transport in the Upper Troposphere and Lower Stratosphere (UTLS). Specifically, we compare partial columns of CO (spanning 147 to 68 hPa) observed by the Microwave Limb Sounder (MLS) and the Atmospheric Chemistry Experiment Fourier Transform Spectrometer (ACE-FTS), as well as in situ observations, with corresponding model data (Wright et al., 2025). Our analysis focuses on zonally resolved seasonal CO patterns, averaged over multiple years, which are largely influenced by seasonal variability in convection. Since chemistry plays a minor role in this study, we employ a simplified CLaMS configuration that includes CO, ozone, and water vapor, with boundary conditions as described by Pommrich et al. (2014).

#### 2 CLaMS adaptive grid with convection

Unlike the more common Eulerian frame of reference, where atmospheric motion is observed from a fixed spatial position, the Lagrangian approach follows individual fluid parcels as they move with the flow. In a numerical model, this distinction means that while the Eulerian grid remains fixed in space, the Lagrangian grid moves along with the flow, materially anchored to it. Thus, the Lagrangian grid, or Lagrangian air parcels (APs), can be understood as massless pivotal points that are intrinsically connected with the flow. Their motion, driven by the flow velocities—either forward or backward in time—defines the so-called Lagrangian trajectories. In both frames of reference, the physical properties at a given grid point represent the mean properties

of the atmosphere within the volume of the grid box surrounding that point. While calculating grid volume in Eulerian models is straightforward, Lagrangian grids require the concept of nearest neighbors and Voronoi volumes (see Appendix A).

Lagrangian transport is inherently less diffusive than Eulerian transport, the latter requiring frequent interpolations on the background grid due to the Courant-Friedrichs-Lewy condition (Courant et al., 1928), which often leads to substantial numerical diffusion, exceeding observed atmospheric diffusion. However, purely Lagrangian trajectories without any mixing represent an unrealistic extreme that does not align with observations (see e.g. Pan et al. (2006)). CLaMS addresses this limitation through an adaptive grid procedure that combines Lagrangian trajectories (i.e., the advective or reversible part of transport) with mixing (i.e., the irreversible part of transport). In this context, "adaptive" means that not only do the positions of the APs within the grid evolve during advective transport along Lagrangian trajectories, but also the number of APs dynamically adjusts based on the flow. This adjustment, always applied between pure trajectory steps in CLaMS, enhances the spatial resolution of the grid when strong flow deformations occur, while simultaneously reorganizing the grid to keep the total number of APs within certain limits. In the following, we revisit the key concepts of the CLaMS adaptive grid, with a focus on its role in representing convection—a critical component of transport between the Earth's surface and the middle and upper troposphere.

#### 2.1 Advection

To describe the 3D position  ${\bf r}$  of an AP in CLaMS, longitude  $\ell$  and latitude  $\varphi$  are used as the horizontal coordinates, while the hybrid potential temperature  $\zeta$  represents the vertical coordinate (Mahowald et al., 2002). The notation  ${\bf r}(\ell,\varphi,\zeta,t)$  denotes the exact position of the AP in space and time. "Hybrid" refers to the combination of the dry potential temperature  $\theta$ , which remains constant during adiabatic motion, and the orography-following coordinate  $\sigma=p/p_s$  (where p is pressure and  $p_s$  is surface pressure). This combination enables the model to resolve transport processes in both the stratosphere and troposphere, where orography plays a role. The transition between  $\sigma$  and  $\theta$  is controlled by the parameter  $\sigma_r=p/p_s$ , set to 0.3 in CLaMS, with pressure p at location p. The Lagrangian trajectory, which quantifies the advective motion of an AP in CLaMS, is determined by numerically integrating the equation of motion:

$$\dot{\mathbf{r}} := \frac{D\mathbf{r}}{Dt} = \mathbf{u}(\mathbf{r}, t), \qquad \mathbf{r}(t_0) = \mathbf{r}_0, \qquad \mathbf{u} = (u, v, \dot{\zeta})$$
(1)

Here, u and v are the horizontal velocities (in m/s), and  $\dot{\zeta}$  is the hybrid vertical velocity (in K/day), composed of the diabatic heating rate  $\dot{\theta}$  and  $\dot{\sigma}$ , the latter being a function of the kinematic vertical velocity in pressure coordinates ( $\dot{p}$ ) and the surface pressure tendency ( $\dot{p}_s$ ) (for details, see Mahowald et al. (2002); Ploeger et al. (2010)). In CLaMS, the Runge-Kutta 4th-order method is used for the numerical integration of eq. (1). The 4D fields u, v,  $\dot{\theta}$ ,  $\dot{\sigma}$ , and  $\dot{p}$  are provided by the reanalysis, here by ERA5 (Hersbach et al., 2020).

Figure 1. Longitude  $(\ell)$ , latitude  $(\varphi)$ , and hybrid potential temperature  $(\zeta)$  describe the spatial positions of air parcels (APs) in CLaMS. The trajectory quantifies the advective motion of an AP in a time- and space-dependent velocity field  $(u, v, \dot{\zeta})$ , which is updated at a frequency of  $1/\Delta t$ , typically with  $\Delta t$  ranging from 1 to 6 hours. This is also the frequency at which unresolved convection—i.e., convection not captured in the velocity fields—is incorporated. Two trajectory examples are shown: a quasi-isentropic trajectory (i.e., almost adiabatic,  $\theta \sim \text{const}$ ) (black), and a strongly diabatic trajectory affected by a convective updraft  $(\Delta \theta)$  (red). The convective updraft lifts the AP from the Earth's surface to a point above the surface  $\theta = \text{const}$ , after which it gradually descends back to the  $\theta = \text{const}$  level. The red dotted points represent the projection of this trajectory onto the  $\theta = \text{const}$  surface.

The black solid line in Fig. 1 shows an example of the time evolution of a typical Lagrangian trajectory calculated from prescribed 3D winds provided by the reanalysis. On timescales of hours to days, atmospheric motion is predominantly isentropic (i.e., with  $\theta \sim$  const), not only in the stratosphere but also in large parts of the free troposphere, far from convective cells. Cross-isentropic motions mainly occur over longer timescales and are primarily driven by radiative imbalances (Krüger et al., 2008; Schoeberl and Dessler, 2011; Legras and Bucci, 2020). The exception is convection, which uplifts air from the Planetary Boundary Layer (PBL) into the free atmosphere, where latent heat release during the condensation and freezing of water vapor plays a dominant role. The red line in Fig. 1 illustrates such a trajectory, featuring a strong diabatic and highly localized (net) updraft  $\Delta\theta$ , followed by subsequent slow, large-scale diabatic subsidence, mainly due to longwave cooling of radiatively active water vapor transported from the main convective outflow (e.g. Poujol and Bony (2024) and references therein). If these convective updrafts are resolved by the meteorological wind data, they appear primarily in the vertical velocity fields  $\dot{\zeta}$ . If not, i.e., if unresolved, the respective updrafts must be parameterized. As described in Konopka et al. (2019, 2022), CLaMS triggers unresolved convection using specific values of either the Convective Available Potential Energy (CAPE) or the moist Brunt-Väisälä frequency,  $N_m$ . The value of  $\Delta\theta$  is derived from the total available latent heat of the water vapor (transforming from gas to ice) in the CLaMS AP within the PBL, as provided by the reanalysis.

#### 2.2 CLaMS grid




The core concept of CLaMS leverages the numerical diffusion that arises from interpolations between APs to parameterize atmospheric mixing, as implemented in the adaptive grid procedure described in previous studies (McKenna et al., 2002b; Konopka et al., 2004, 2012). The numerical horizontal and vertical diffusion,  $D_h$  and  $D_v$ , associated with interpolations between two neighboring APs, scale as  $\sim r^2/\Delta t$  and  $\Delta z^2/\Delta t$ , where r and  $\Delta z$  represent the geometric horizontal and vertical mean separation between the APs, and  $1/\Delta t$  represents the interpolation frequency (see, e.g., Press et al. (2007), chapter 20.2: Diffusive Initial Value Problems). Consequently, the ratio  $D_h/D_v = \alpha^2$ , where  $\alpha = r/\Delta z$ , quantifies the atmospheric aspect ratio, reflecting the ratio of atmospheric variability scales (Haynes and Anglade, 1997). Since this relation aligns with observations (Balluch and Haynes, 1997), it suggests that simple interpolations and their associated numerical diffusion can effectively mimic atmospheric diffusion. Thus, the parameters r and  $\Delta z$ , which are generally time- and space-dependent, can be used to define the Lagrangian grid. The distances between neighboring APs—i.e., the grid itself—and the frequency of adaptive re-gridding are therefore critical for accurately capturing the diffusive properties of the entire atmosphere.

CLaMS relies on forward Lagrangian transport and mixing among APs that cover the free atmosphere, from the top of the PBL up to the mesosphere, following the concept described by Konopka et al. (2012). In our standard configuration, almost 2 million APs are distributed from the surface to the mesosphere. The grid is defined by two key parameters: entropy density s and the dry Brunt-Väisälä frequency  $N_d$ , from which the grid parameters r (horizontal separation) and  $\Delta z$  (vertical separation) are derived. First, we ensure that each AP represents the same amount of entropy s, which guarantees a consistent representation of mixing processes across different atmospheric regions, as mixing is an irreversible process that increases entropy. In a materially closed system, the specific internal energy depends solely on the specific entropy s, so APs with the same entropy also share the same internal energy. Second, we assume that the aspect ratio  $\alpha \sim N_d$ . This assumption helps shape the adaptive grid to reflect the atmosphere's varying mixing properties due to stability differences, with a well-mixed tropopause and a stably stratified stratosphere.

**Figure 2.** CLaMS grid extending from the Earth's surface to the mesosphere (grey points). The density of air parcels is highest around the tropical tropopause. Four idealized APs are shown: three in the free atmosphere (blue) and one in the Planetary Boundary Layer (PBL, red). While the APs in the free atmosphere are generated following the 1D entropy-static stability concept described in Konopka et al. (2012), the generation of APs in the PBL is decoupled from this procedure to ensure sufficiently high resolution, in both time and space, to resolve small-scale processes such as convective updrafts not represented in the meteorological velocity fields or various sources of tracers.

As air mass density  $(\rho)$  decreases and dry potential temperature  $(\theta)$  increases with altitude, the entropy density  $(s \sim \rho \ln \theta)$  reaches a maximum—using an idealized 1D US standard atmosphere—near the tropical tropopause. Consequently, an even distribution of s across all APs results in the highest grid point density around this region (see Fig. 2). Given that  $N_d$  peaks in the lower stratosphere, and following our assumption that  $\alpha \sim N_d$ , the corresponding APs exhibit large horizontal extensions and reduced vertical separations, consistent with the observed large values of  $\alpha$  in this region (Balluch and Haynes, 1997). In contrast, APs in the middle troposphere and upper stratosphere exhibit significantly smaller  $\alpha$  values (cf. the ratios of the horizontal to vertical edges of the blue rectangles in Fig. 2). This relationship between  $N_d$  and  $\alpha$  allows us to mimic the diffusive properties of the atmosphere, which is characterized by a well-mixed troposphere and strongly suppressed vertical mixing in the stratosphere, where static stability  $N_d$  is very high (see e.g. Grise et al. (2010)). Using idealized 1D profiles of entropy s and Brunt-Väisälä frequency  $N_d$  from the US standard atmosphere, the CLaMS grid, defined by  $r(\zeta)$  and  $\Delta z(\zeta)$  (or rather  $\Delta \zeta(\zeta)$ ), is generated with  $r = r_0$  at  $\zeta = 380$  K (tropical tropopause), based on  $\zeta$  values provided by the US standard atmosphere with  $\sigma_r = 0.3$  (Konopka et al., 2012). By varying  $r_0$ , different model resolutions can be chosen, with the standard configuration set by  $r_0 = 100$  km (see Appendix B).

While this grid design, based on simple thermodynamics, provides a good approximation for the free atmosphere, it becomes less applicable near the Earth's surface, where transport is strongly influenced by boundary effects such as friction and convection. In this region, interactions with the Earth's surface and PBL dynamics introduce additional small-scale sources of tracers and convective updrafts from small areas (~1 square kilometer and below), which are critical for accurate modeling of atmospheric transport. Examples of such small-scale sources include localized emissions from urban centers, industrial facilities, and biomass burning events, as well as biogenic emissions from vegetation. To account for these boundary effects, additional information must be integrated into the model. In information theory, Shannon entropy is sometimes used to quantify these additional entropy fluxes (Shannon, 1948). In our simplified approach, we decouple the APs in the PBL from those in the free atmosphere and significantly increase the resolution within the PBL by reducing the volume of the respective APs. This enhanced resolution more accurately captures the finer-scale processes characteristic of near-surface atmospheric dynamics.

After initializing the grid as described above, the advective trajectories transport the APs to their new positions. If necessary, the chemistry module updates the chemical composition of each AP. The advective time step,  $\Delta t$ , is typically set to 24 hours, though shorter values between 6 and 24 hours may also be used (note that the internal time step in the Runge-Kutta scheme for trajectory calculation is much smaller). Once the advection step is completed, the CLaMS adaptive grid procedure is applied to parametrize mixing and rearrange the grid.

## 2.3 Adaptive grid procedure








This procedure consists of two parts: the insertion of new APs when the nearest neighbors (NNs) of the considered AP move too far apart during the time step  $\Delta t$  (deformation-induced regridding) and the merging of too-close NNs to maintain the total number of APs within a prescribed range. The key numerical challenge in both parts is the computationally expensive calculation of the NNs, which, in the current CLaMS version, is effectively performed in 2D using Delaunay triangulation (see Appendix A). Since isentropic mixing typically occurs in relatively thin, quasi-2D atmospheric layers, this approach is physically well-justified. The adaptive grid procedure is thus performed layer by layer, dividing the full atmosphere into nearly isentropic layers i, with the altitude-dependent thickness  $\Delta \zeta_i$  defined by the entropy s and the static stability  $N_d$  profiles introduced in the previous subsection.

As shown in Fig. 3, deformation-induced regridding, which parameterizes isentropic mixing, considers only those APs whose final vertical positions (i.e., after the advection step) are within the layer  $\Delta \zeta_i$  and whose initial vertical positions are within a small but prescribed range  $\Delta \zeta$ , ensuring an isentropic approximation. An AP affected by convective uplift from the boundary layer during the advection time step, as exemplified by the red AP, is excluded from this part of the adaptation procedure because it significantly violates the isentropic approximation.

Consider an AP and its position before being advected by a Lagrangian trajectory. We calculate its NNs using the horizontal coordinates prior to advection, referred to as "old" coordinates, to distinguish them from the "current" coordinates after the advection step. Deformation-induced regridding means that new APs are inserted between P and its NNs (calculated with the old coordinates) if their separations r after the advection increase beyond a prescribed value  $r_+$  (in our example, this applies to segments PA, PB, and PC), with relevant quantities interpolated at these new positions from their respective neighbors

Figure 3. Layerwise adaptive grid procedure in CLaMS with unresolved convection (red AP). After the advection step, all APs are grouped into horizontal layers with thickness  $\Delta\zeta_i$ . For example, this applies to AP P and the other seven APs in the right ellipse (for better visibility, the layer  $\Delta\zeta_i$  is shifted downwards, with dashed lines marking the respective vertical positions of all considered APs). These seven APs are considered because they are the (old) nearest neighbors (NNs) of P before the advection step (left ellipse), i.e., their horizontal distance to P was  $\sim r$ , where r represents the mean separation of APs in layer  $\Delta\zeta_i$ . Since the vertical positions of these APs before the advection step lie within a small but prescribed range  $\Delta\zeta$  (typically  $\Delta\zeta \approx \Delta\zeta_i$ ), their respective advection is considered quasi-isentropic. The deformation-induced mixing (the first part of grid adaptation) inserts new APs if the distance between NNs becomes larger than  $r_+$  (as is the case for segments PA, PB, and PC, marked by green crosses). In the second part of grid adaptation, APs are merged if their separation from their (current) NNs is smaller than  $r_-$ . Only during this part can APs that have moved from outside of  $\Delta\zeta$  into  $\Delta\zeta_i$ , such as the red AP denoting convection, undergo interpolation due to merging.

(for details on interpolation, see Appendix C). Note that r before the advection represents the mean horizontal distance in the considered layer  $\Delta \zeta_i$ . We refer to this part of grid adaptation as deformation-induced regridding, as the drifting apart of neighboring APs is mainly caused by vertical shear and horizontal strain, typically understood as precursors to mixing in stably stratified flow.

Since deformation-induced regridding increases the total number of APs (in our example, by 3), the merging step of the adaptive grid procedure is applied to balance the insertion of new APs and maintain the total number within a prescribed range. For this, a renewed 2D triangulation is performed using the current positions of the APs, and air parcels are merged if their distance is smaller than a prescribed threshold  $r_{-}$ . In this process, the red point introduced by convective uplift may also be merged with other APs. Note that the advection and convection time steps are decoupled in this approach: the advection time step is typically 24 hours, while the convection time step corresponds to the maximum frequency of the available reanalysis data (every 6 hours here, though 1-hour data are also available). The convection time step is also typically the interval when the lower boundary, defining species values in the PBL, is updated with new data.

The choice of parameters  $r_{\pm}$  is somewhat arbitrary. While  $r_{+}$  depends on  $\Delta t$  and should quantify the strength of critical deformations that trigger mixing, the approach where  $r_{+} = r \exp(\lambda \Delta t)$  (with r as the horizontal mean separation in layer i and  $\lambda$  the Lyapunov exponent) has proven effective for triggering AP insertion. The choice of  $r_{-}$  is more pragmatic, motivated by the need to confine the total number of APs and prevent unlimited growth in the number generated during insertion. We use  $r_{-} = \epsilon r \exp(-\lambda \Delta t)$ , with  $\epsilon$  ranging between 0.3 and 1, for  $\Delta t = 6$  and 24 hours, respectively. This empirical choice of  $\epsilon$  ensures that the total number of APs varies by less than 40%. For typical values of the parameters  $\Delta t$ ,  $\lambda$ , and  $\epsilon$ , see Table 1.

#### 180 2.4 A Few Remarks






Layerwise adaptive grid and hybrid potential temperature. Ideally, the layers in the adaptive grid procedure should be parallel to the levels of the surfaces of constant potential temperature  $\theta$  to ensure, ensuring "nearly" isentropic interpolations within the layers. Using When using the hybrid potential temperature  $\zeta$  with  $\sigma_r = 0.3$  and far away from orography, this is provided above  $\sim \sigma_r = 0.3$ , this condition is generally satisfied above about 300hPa. In the region below that level hPa, away from strong orography. In regions influenced by orography, and up to  $\frac{100 \text{ hPain regions influenced by orographyapproximately 200 hPa}, \zeta$  surfaces eross the potential temperature levels. intersect the isentropic levels. The use of a hybrid  $\zeta$  coordinate is particularly advantageous in this context, as it allows the model to better capture adiabatic motions such as gravity waves, which are less well handled by purely kinematic schemes. However, in regions of strong orographic forcing, this problem is still not completely solved. Increasing the parameter  $\sigma_r$  would make the model more diabatic, as the respective vertical velocities would then be calculated from the diabatic heating rates  $\dot{\theta}$  rather than from the kinematic vertical velocity  $\dot{p}$  (see Appendix A in Tao et al. (2018)). The optimal parameter choice still requires further investigation.

2D Delaunay triangulation. The layerwise adaptive grid procedure uses 2D Delaunay triangulation to calculate the NNs of all APs in the layer, both before and after the advection step. 2D triangulation projects all APs onto a sphere and, in this way, neglects vertical separation between the APs. Since potential NNs across the boundary between the layers are neglected, this approximation is slightly weakened by considering alternating layer thickness every second advective step  $\Delta t$  (Konopka et al.,

2004). 2D Delaunay triangulation on a sphere is not only numerically more efficient but also conceptually simpler, as there are no boundaries on a sphere.

Mass conservation. As in Eulerian models, a known concentration or mixing ratio, together with the volume of the grid box, provides the mass, which can also be determined for the Lagrangian grid, although the Eulerian grid cell must be replaced by the Voronoi volume (see Appendix A). Thus, the mass of a species, such as a chemically passive tracer, can be calculated both locally and for the entire atmospheric domain. It is important to note that interpolations within the CLaMS adaptive grid do not inherently conserve mass (see Appendix C). A few points are worth mentioning here. Regarding global mass conservation within the model, the total mass of each chemically passive species in the resolved atmospheric domain should balance with the fluxes into and out of this domain. This is trivially true in stationary (i.e., time-independent) atmospheric states, where all concentrations and fluxes are constant, and consequently, the system's mass remains unchanged. Thus, in cases where chemical production is roughly balanced by chemical destruction, and the atmospheric composition is solely driven by seasonality, mass conservation is inherently preserved in CLaMS as the model remains close to a stationary state.

Alternatively, Lagrangian grids can be formulated with a constant number of APs, each containing the same mass. While such grids maintain a constant total mass by design, they face significant challenges. First, the number of APs decreases with altitude as atmospheric mass declines exponentially with height. Consequently, fewer APs are available in the upper atmosphere, potentially insufficient to accurately capture relevant physical processes. Second, if the 3D velocity fields do not precisely satisfy the continuity equation, the initially uniform distribution of APs can become distorted, leading to local violations of mass conservation. For example, in such a model, all APs could migrate into one hemisphere, compromising the model's physical accuracy. The CLaMS adaptive grid addresses these issues by inserting new APs to fill potential gaps or reducing the number of APs in overly clustered regions. This approach ensures that the model maintains high-quality transport even in the presence of these challenges.

#### 3 Adaptive Grid for Different CLaMS Versions







To visualize the performance of the CLaMS adaptive grid, we compare the standard configuration of CLaMS-3.0 with CLaMS-1.0 (Pommrich et al., 2014) and CLaMS-2.0 (Konopka et al., 2022). The key differences between these versions are summarized in Table 1. Three separate CLaMS runs are initialized on 01 January 1979 and span 44 years, reaching the end of 2023. The initial grid of APs is generated as described in the previous section, using the 1D entropy-static stability concept (Konopka et al., 2012). The nominal horizontal resolution is set to  $r_0 = 100$  km, with an aspect ratio of  $\alpha = 250$  at  $\zeta = 380$  K. The thickness and horizontal mean distance between the APs in the planetary boundary layer (PBL) vary slightly between the versions, as outlined in Table 1, with  $\Delta \zeta$  ranging from 100 to 140 K. The choice of  $\Delta \zeta = 140$  K corresponds to a geometric thickness varying between 1.6 and 2.2 km.

During the simulation, after each advective time step  $\Delta t$ , the adaptive grid procedure adjusts the resolution of the initial grid based on flow deformations. Typically, the total number of APs grows by 30-40%—from approximately 1.3, 1.4, and 1.8 million for CLaMS-1.0, 2.0, and 3.0, respectively—before stabilizing throughout the integration. Fig. 4 illustrates the time

| Version | PBL                                | Adaptive grid                                                            | Convection                       | Interpolation | Reference             |
|---------|------------------------------------|--------------------------------------------------------------------------|----------------------------------|---------------|-----------------------|
| 1.0     | $\Delta \zeta$ =100 K, $r$ =110 km | $\Delta t$ =24 h, $\lambda$ = 1.5 day <sup>-1</sup> , $\epsilon$ = 1.0   | no                               | linear        | Pommrich et al., 2014 |
| 2.0     | $\Delta \zeta$ =140 K, $r$ =90 km  | $\Delta t = 6 \text{ h}, \lambda = 4.0 \text{ day}^{-1}, \epsilon = 0.3$ | $N_m$ , $\Delta t = 6 \text{ h}$ | linear        | Konopka et al., 2022  |
| 3.0     | $\Delta \zeta$ =140 K, $r$ =40 km  | $\Delta t$ =24 h, $\lambda$ = 1.5 day $^{-1}$ , $\epsilon$ = 1.0         | CAPE, $\Delta t = 6 \text{ h}$   | weighted      | this paper            |

Table 1. Key differences among the CLaMS versions. The Planetary Boundary Layer (PBL) is characterized by its thickness ( $\Delta\zeta$ ) and the mean distance between air parcels (APs), denoted by r. The adaptive grid procedure is defined by its frequency ( $1/\Delta t$ ), the Lyapunov exponent ( $\lambda$ ), and the parameter  $\epsilon$ , which collectively determine the criteria for the insertion and merging of APs. The parameterization of convective uplifts, representing unresolved convection, is triggered by either the moist Brunt-Väisälä frequency ( $N_m$ ) or the Convective Available Potential Energy (CAPE), with critical values set to 0 for  $N_m$  in CLaMS-2.0 and 800 J/kg for CAPE in CLaMS-3.0. In CLaMS-2.0, the adaptive and convective time steps are synchronized, whereas in CLaMS-3.0, they are decoupled. Furthermore, CLaMS-3.0 uses weighted interpolations during the insertion and merging of APs (see Appendix C), unlike the simple linear interpolations employed in earlier versions.

evolution of globally integrated profiles for the total number of APs in CLaMS-3.0 (left panel) and the differences in such profiles for all three CLaMS versions, as exemplified on 31 August 2017 (right panel). The gray line in Fig. 4 represents an idealized number density distribution, assuming that all APs have the same mass. The CLaMS distributions generally follow this mass-scaling between 380 and 450 K but intentionally deviate from it above 500 K, where more APs are used to resolve the atmospheric structure in this region.






After an initial spin-up period of approximately two months, the total number of APs in CLaMS-3.0 stabilizes at around 2.4 million, with little change thereafter (left panel). On 31 August 2017, a representative snapshot day, three distinct peaks in AP density are observed around 600 K, 340 K, and between 140 and 260 K. These peaks are present across all three versions of CLaMS (right panel), with CLaMS-3.0 extending to the Earth's surface in contrast to the previous CLaMS versions. Fig. 5 further illustrates these features, showing zonal means of AP density (left) and mixing intensity (right), using an area-preserving latitude grid horizontally and the initial ζ-grid vertically. The three maxima in AP density correspond to the locations of jets and the main convective outflow in the tropics. Specifically, they highlight increased AP density around the polar vortex over Antarctica and the subtropical jets, as well as in the tropical main convective outflow. The mixing intensity reflects the percentage of APs influenced by regridding and numerical diffusion, with higher values near the jets indicating more active mixing. Additionally, a distinct feature in the tropics around 500 K is driven by the Quasi-Biennial Oscillation (QBO), where strong wind shear between easterlies above and weaker westerlies below causes flow deformations and drives mixing. Strong convective activity below the tropical tropopause moves APs from the PBL into the Tropical Tropopause Layer (TTL), enhancing AP density and triggering regridding in the adaptive grid procedure.

Similar features are observed near the tropospheric jets between 70 and 250 K. In CLaMS-3.0, higher AP density in the PBL further amplifies this effect. An asymmetry in stratospheric AP distributions between the Southern Hemisphere (SH) and Northern Hemisphere (NH) is evident, particularly in the CLaMS-1.0 and 3.0 configurations, reflecting the dynamic contrast between the vigorous winds of the winter SH and the calmer winds of the summer NH, sometimes described as "solid body

**Figure 4.** Vertical profiles of the globally integrated total number of APs, plotted as step-wise functions with step heights corresponding to the layer thicknesses. Left: Time evolution for the CLaMS-3.0 configuration. Right: Differences between the CLaMS versions on 31 August 2017. The gray line represents a distribution scaled by atmospheric density, assuming that each AP has the same mass.

rotation". In contrast, this asymmetry is less pronounced in the CLaMS-2.0 configuration, which appears to be more affected by numerical noise, with a weaker representation of the QBO effect. This discrepancy arises from the higher frequency of grid adaptation in CLaMS-2.0 (6 hours versus 24 hours in CLaMS-3.0), making the detection of deformations more sensitive to numerical noise during the re-gridding. Despite the improved representation of tropospheric transport in CLaMS-2.0 compared to CLaMS-1.0 (Konopka et al., 2022), we exclude CLaMS-2.0 from further discussions and focus henceforth on comparing CLaMS-3.0 with CLaMS-1.0.

#### 4 Model Evaluation with CO Satellite and In Situ Data



The most significant updates in CLaMS-3.0 pertain to the representation of upward transport from the PBL to the UTLS region. Carbon monoxide (CO) serves as an excellent tracer for validating the model's troposphere-to-stratosphere transport due to its relatively short lifetime of around 3 months, which enables the tracing of distinct signatures of convective transport (Park et al., 2009; von Hobe et al., 2021; Pan et al., 2024). Moreover, CO benefits from extensive satellite-based observations (Hegglin et al., 2021) and numerous local in situ measurements, which can be utilized both for defining the CLaMS lower boundary and for validating the transport and chemistry implemented in the model.

Figure 5. Adaptive grid properties for CLaMS-1.0, 2.0, and 3.0, shown for 31 August 2017 as an example. Zonal means of AP number density (n, left) and mixing intensity (Mix. Int., right) are presented, calculated per grid box defined by an area-preserving latitude grid and the  $\zeta$ -grid generated at initialization. The mixing intensity quantifies the percentage of APs affected by the adaptive grid procedure. The vertical positions of the layers, total number of levels, and total number of APs are indicated on the right side of each panel. Solid and dashed lines represent westerlies and easterlies, respectively, while the thick white line represents the tropopause.

In particular, satellite observations from the Measurement of Pollution in the Troposphere (MOPITT) and the Atmospheric Infrared Sounder (AIRS) provide suitable lower boundary conditions for the model, capturing both the spatial and temporal variability of CO with high accuracy. Using back- and forward-trajectory calculations, we grid the MOPITT (2001–2012, version 4) and AIRS (2013–2020, version 6) data at a latitude-longitude-time resolution of  $2^{\circ}$ ,  $6^{\circ}$ , and 5 days, respectively, at  $\zeta = 200$  K (corresponding to  $\sim 500$  hPa) (Pommrich et al., 2014). These particular time periods were chosen to maintain continuity with previous studies and datasets used in earlier phases of the model's development, ensuring consistency across analyses. This approach establishes the CLaMS lower boundary by overwriting all CLaMS APs with  $\zeta$  values below 200 K after each advective time step,  $\Delta t$ .

## 4.1 Comparison of CLaMS Model to MLS and ACE-FTS Satellite Observations







To evaluate CLaMS-3.0, we use Microwave Limb Sounder (MLS) version 5.1 (Livesey et al., 2020) and Atmospheric Chemistry Experiment Fourier Transform Spectrometer (ACE-FTS) version 5.2 (Sheese et al., 2015; Boone et al., 2023; Sheese and Walker, 2023) satellite observations spanning the period 2005–2020. MLS, a limb-sounding instrument, provides extensive global coverage with dense daily observations but has a coarser vertical resolution of approximately 5–5.5 km for CO (Santee et al., 2017). In contrast, ACE-FTS, based on solar occultation measurements, offers higher vertical resolution (1–3 km for CO; (Hegglin et al., 2009)) but with much sparser spatial and temporal sampling, constrained to specific sunrise and sunset conditions along its orbit.

We analyze partial columns of CO (XCO), calculated in mg/m², for both satellites and CLaMS across the pressure range from 146.8 to 68.1 hPa, as discussed in Wright et al. (2025). This metric approximately captures the entire UTLS region in both tropical and extratropical zones, providing a convenient integral quantity to study the zonally resolved seasonality of CO. While the high vertical resolution of ACE-FTS allows for direct comparison with CLaMS, applying the MLS averaging kernels to the higher-vertical-resolution model data would be necessary for direct comparisons with MLS profiles. However, since the CLaMS partial columns calculated with and without applying the MLS averaging kernel differ by less than 5% (see Appendix D), we restrict our comparisons to simple partial columns derived from both MLS and CLaMS. The zonally resolved annual mean climatologies for 2005–2020 for both satellites and CLaMS simulations, along with their zonal means, are shown in Fig. 6.

First, as ACE-FTS and MLS climatologies act as references, it should be noted that they differ both in absolute values and spatial patterns, with MLS values being approximately 10% higher in the tropics and more than 100% higher poleward of  $50^{\circ}$ . As discussed in Livesey et al. (2020), a substantial high bias (factor of  $\sim$ 2) in version 2 (v2) MLS CO, as revealed through comparisons with various in situ CO datasets, has been essentially eliminated in version 5 (v5). However, the quoted systematic uncertainties in v5 MLS CO remain unchanged from those in version 4 (v4), meaning the  $2\sigma$  values provided for the UTLS in the study by Santee et al. (2017) are still valid. These uncertainties, which are rather large ( $\pm$ 30 ppbv +  $\pm$ 30%), may be sufficient to encompass the magnitude of the biases identified in the comparisons with ACE. On the other hand, zonal asymmetries compare well between the two climatologies, with one notable exception: the region over northwest Australia,

**Figure 6.** Annual mean climatology of CO partial columns (XCO) between 147 and 68 hPa, derived from MLS and ACE-FTS data for the period 2005–2020 (first two panels on the left), and from CLaMS-1.0, CLaMS-3.0, and CLaMS-3.0 with enhanced CO at the lower boundary (CLaMS-3.0/sens) (right). The respective zonal means are shown in the bottom left panel.

where high CO values are visible in ACE-FTS but not in the MLS climatology. Here, sampling biases in ACE-FTS due to its non-uniform coverage cannot be ruled out (Kloss et al., 2019). Such differences make model validation somewhat challenging.

Our first robust finding is that the observed CO partial column values are significantly underestimated by CLaMS-1.0, while CLaMS-3.0 shows notable improvements. However, the model values of XCO remain smaller than those from the satellite instruments. As can be deduced from the comparison of zonal means (Fig. 6, bottom left), CLaMS-3.0 underestimates XCO from ACE-FTS by around 15% in the tropics and only by a few percent elsewhere. The low bias relative to MLS is larger, especially in the extratropics, similar to the differences between ACE-FTS and MLS.






As vertical transport from the PBL to the UTLS might still not be fully captured in CLaMS-3.0, another potential limitation is the relatively coarse spatial resolution (2° and 6° in latitude and longitude) and temporal resolution (every 5 days) of the lower boundary derived from MOPITT and AIRS. These observations are smoothed by their averaging kernels, potentially leading to an underestimation of CO values. To assess this effect, we performed a sensitivity study (CLaMS-3.0 with enhanced CO) by scaling the lower boundary with factors of 1.5 and 1.25 for the MOPITT and AIRS datasets, respectively, reflecting the generally smaller CO values in MOPITT. Although somewhat arbitrary, these scaling factors were chosen to adjust the lower boundary to upper-range values typically observed in the PBL (see also Pommrich et al. (2014); Vogel et al. (2016)). In the tropics, this adjustment reversed the low bias in CLaMS XCO to a slight high bias relative to MLS (see Fig. 6, bottom left panel). However, a significant low bias still persists in the extratropics relative to MLS.

Before introducing in situ data to investigate this discrepancy further, we compare seasonal anomalies between CLaMS-3.0 and MLS, which offers denser spatial coverage compared to ACE-FTS. The results are shown in Fig. 7, with panels from top to bottom representing DJF, MAM, JJA, and SON.

To assess the agreement between observed CO anomalies and those simulated by CLaMS-1.0 and CLaMS-3.0, we computed spatial correlation coefficients for each season using data from both MLS and ACE-FTS. The results are presented in Table 2. Notably, CLaMS-3.0 consistently shows higher correlation coefficients across all seasons compared to CLaMS-1.0, indicating

**Table 2.** Seasonal Spatial Correlation Coefficients between Observed CO Anomalies (MLS/ACE-FTS) and Model Simulations (CLaMS-1.0 and CLaMS-3.0) as shown in Fig. 7

| Season    | DJF       | MAM       | JJA       | SON       | ALL       |
|-----------|-----------|-----------|-----------|-----------|-----------|
| CLaMS-1.0 | 0.55/0.00 | 0.44/0.58 | 0.94/0.43 | 0.69/0.48 | 0.67/0.63 |
| CLaMS-3.0 | 0.82/0.13 | 0.91/0.45 | 0.97/0.42 | 0.90/0.49 | 0.91/0.64 |

a more accurate representation of the spatial distribution of CO anomalies. In particular, positive XCO anomalies over Africa during DJF and MAM, as well as over South America during DJF and SON, are more pronounced in CLaMS-3.0. Additionally, the enhanced influence of the Asian summer monsoon during JJA is better captured in CLaMS-3.0 compared to CLaMS-1.0. The superior spatial coverage of MLS data compared to ACE-FTS likely contributes to the higher correlation coefficients observed with CLaMS-3.0. This discrepancy is particularly evident during SON over northwest Australia, where elevated CO values are detected by ACE-FTS but are absent in both MLS climatology and our CLaMS-3.0 simulations.

**Figure 7.** Seasonal anomalies (DJF, MAM, JJA, and SON) of the CO partial column,  $\Delta$ XCO, derived from the 2005–2020 climatology for MLS (left), CLaMS-1.0 (middle), and CLaMS-3.0 (right). The anomalies are calculated relative to the zonal mean for the respective season.

Sensitivity studies for 2006, a year when these differences are most pronounced, show that incorporating a parameterization for unresolved convection is critical for reproducing such spatial anomalies (Table 3, second row). However, the choice of

**Table 3.** Sensitivity analysis for 2006: Spatial correlation coefficients between MLS and different CLaMS-3.0 configurations. The first row represents the reference case, where all sensitivities are switched on (1 = enabled, 0 = disabled). The other rows show the impact on correlation when each sensitivity is disabled.

| Unres. conv. | High res. PBL | Low. boundary | DJF  | MAM  | JJA  | SON  | ALL  |
|--------------|---------------|---------------|------|------|------|------|------|
| 1            | 1             | 1             | 0.72 | 0.87 | 0.94 | 0.81 | 0.85 |
| 0            | 1             | 1             | 0.42 | 0.29 | 0.91 | 0.61 | 0.57 |
| 1            | 0             | 1             | 0.64 | 0.81 | 0.93 | 0.77 | 0.79 |
| 1            | 1             | 0             | 0.48 | 0.50 | 0.89 | 0.64 | 0.60 |

parameterization type (CAPE-based or the moist Brunt-Väisälä frequency  $N_m$ ) has a smaller impact (not shown). The increased density of APs (40 km versus 110 km as in CLaMS-1.0) plays a much weaker role in representing anomalies (Table 3, third row) but is more crucial in enhancing the absolute values of CO columns (not shown).

Interestingly, replacing CO with an E90-tracer-based diagnostic (Prather et al., 2011; Abalos et al., 2017), which mimics CO behavior through a 90-day decay, and substituting the satellite-based lower boundary with a uniform E90 tracer value of 150 ppbv globally, also shows some potential to reproduce the anomalies (Table 3, last row). This highlights the relative importance of accurately representing vertical transport over the specific details of the lower boundary conditions. Moreover, the relatively high correlation with the E90 tracer further motivates numerical experiments with such an idealized tracer, as they are simpler in design and do not require a lower boundary condition. This simplification makes it easier to optimize transport schemes, for example, in climate models.

In Appendix E, we complement our validation of the UTLS model representation by analyzing ozone partial columns from both CLaMS-3.0 simulations and MLS observations. As ozone is predominantly a stratospheric tracer, unlike carbon monoxide (CO), this comparison primarily assesses the model's performance in the stratospheric part of the UTLS. CLaMS-3.0 reproduces the climatological seasonality and longitudinal structure of MLS-derived ozone remarkably well, with some improvement over CLaMS-1.0. Notably, the model now captures more pronounced ozone minima over the Maritime Continent, reflecting improved convective lofting of ozone-poor air — consistent with the CO-based diagnostics in Sect. 4. This confirms the continued ability of the model to accurately represent stratospheric transport processes (see, e.g., Konopka et al., 2010; Yan et al., 2018)

# 4.2 Comparison with In Situ Data






÷

To further investigate the discrepancies in XCO absolute values, especially the high bias of MLS relative to ACE-FTS and CLaMS-3.0 poleward of 50 degrees, we incorporate in situ airborne observations (Fig. 8) from several campaigns. The REC-ONCILE (Reconciliation of Essential Process Parameters for an Enhanced Prediction of Arctic Stratospheric Ozone Loss and Climate Effects) campaign was carried out in 2010 onboard the M-55 Geophysica aircraft, based in Kiruna, Sweden (von

Hobe et al., 2013). The StratoClim (Stratospheric Climate Links with Emphasis on the Upper Troposphere and Lower Stratosphere) campaign was conducted in 2017, also onboard the M-55 Geophysica aircraft, based in Kathmandu, Nepal (von Hobe et al., 2021). The WISE (Wave-driven Isentropic Exchange) campaign, also in 2017, operated from Shannon, Ireland, using the HALO (High Altitude and Long Range Research Aircraft) (Lauther et al., 2021). The ACCLIP (Asian Summer Monsoon Chemical and Climate Impact Project) campaign took place in 2022, based in Osan, South Korea, using the GV (Gulfstream V) and WB-57 aircraft (Pan et al., 2024). The PHILEAS (Probing High Latitude Export of Air from the Asian Summer Monsoon) campaign was conducted in August and September 2023, with most flights operating from Anchorage, Alaska, onboard HALO.

The campaigns, the geographic positions of their bases, the months covered by the observations, and the longitude-latitude ranges assumed for the comparison with the MLS and ACE-FTS climatologies are listed in Table 4. The CO datasets collected during these campaigns are shown in Fig. 8, using log-pressure altitude as the vertical coordinate, and are compared with the respective profiles derived from the MLS and ACE-FTS climatologies, as well as CLaMS simulations. The comparison reveals that, in agreement with our discussion in the previous subsection, MLS profiles consistently show higher values than ACE-FTS profiles. For instance, MLS XCO values exceed those from ACE-FTS by more than 100% in the polar region (RECONCILE), with the in situ mean profile exhibiting better agreement with ACE-FTS observations than with MLS observations. In other campaigns—except for WISE—particularly above 10 km, the mean in situ observations generally agree better with MLS profiles.



**Table 4.** Summary of campaigns, including the geographic base, months covered, latitude-longitude ranges used in the MLS and ACE-FTS climatologies for comparison with in situ data (Fig. 8), and CO instruments utilized: COLD/COLD2 (Carbon monoxide Laser Detector) (Viciani et al., 2008, 2018), UMAQS (University of Mainz Airborne QCL-based Spectrometer) (Krasauskas et al., 2021), and Aerodyne-CO (Aerodyne Quantum Cascade Laser Absorption Spectrometer) (McManus et al., 2010; ACC, 2023).

| Campaign   | Geographic Base   | <b>Months Covered</b> | Latitude Range | Longitude Range | CO Instrument     |
|------------|-------------------|-----------------------|----------------|-----------------|-------------------|
| RECONCILE  | Kiruna, Sweden    | Jan–Mar 2010          | 65°N–90°N      | 0°-45°E         | COLD              |
| StratoClim | Kathmandu, Nepal  | Jul-Aug 2017          | 10°N–40°N      | 70°E–100°E      | COLD              |
| WISE       | Shannon, Ireland  | Sep-Oct 2017          | 30°N-60°N      | 10°W–30°E       | UMAQS             |
| ACCLIP     | Osan, South Korea | Jul-Sep 2022          | 20°N-50°N      | 110°E–150°E     | COLD2/Aerodyne-CO |
| PHILEAS    | Anchorage, Alaska | Aug-Sep 2023          | 50°N-80°N      | 140°W–170°E     | UMAQS             |

The largest deviation between the in situ data and satellite observations was observed during the ACCLIP campaign, likely due to the sampling of highly polluted Asian monsoon air and the strong La Niña influence discussed in Pan et al. (2024), which makes the climatologies unrepresentative for this campaign. CLaMS-3.0 simulations generally fall within the spread of in situ observations (within the  $1\sigma$  range around the mean profiles) but tend to slightly underestimate the observed values. The best agreement is achieved when using the enhanced lower boundary configuration, except for the StratoClim comparison, as relatively clean air was sampled within the Asian monsoon anticyclone during that campaign. Only CO observations onboard the Geophysica (RECONCILE and StratoClim) and WB-57 aircraft (ACCLIP) cover the full range of XCO considered in this

Figure 8. Observed in situ CO profiles from various campaigns in the Northern Hemisphere (gray points represent all local flights of the respective aircraft) and their mean values (black thick points) calculated over bins defined by the MLS pressure levels, with the respective  $2\sigma$  standard deviations. The profiles are plotted as a function of log-pressure altitude, using  $H=7\,\mathrm{km}$  and a reference pressure of 1000 hPa. The corresponding mean profiles from CLaMS simulations are also shown (see legend). The MLS and ACE-FTS profiles represent mean values and their standard deviations over the months and regions covered by the respective campaigns. The thick black lines on the right y-axes indicate the partial column range of XCO, spanning 147 to 68 hPa, calculated for the MLS and ACE-FTS partial columns.

study. However, only the RECONCILE observations clearly show better agreement with ACE-FTS than with MLS, further confirming the high bias of MLS CO measurements northward of 50°N.

## 5 Conclusions







This study presents an updated and improved version of the Chemical Lagrangian Model of the Stratosphere (CLaMS-3.0/MESSy), featuring key advancements in the representation of transport processes from the Planetary Boundary Layer (PBL) to the Upper Troposphere and Lower Stratosphere (UTLS) region. By enhancing spatial resolution in the PBL and decoupling the frequency of the adaptive grid procedure from the convection parameterization, CLaMS-3.0 more accurately captures the complex interplay between small-scale convective uplift and large-scale flow dynamics. These changes allow for better representation of critical transport pathways and mixing processes, especially for tracers like carbon monoxide (CO).

Using CO as a tracer, we evaluate CLaMS-3.0 against Microwave Limb Sounder (MLS) and Atmospheric Chemistry Experiment Fourier Transform Spectrometer (ACE-FTS) satellite observations, as well as airborne in situ measurements from multiple campaigns. Including a parameterization for unresolved convection significantly improves the representation of CO anomalies in the UTLS region, defined here as the partial columns between 147 and 68 hPa. Seasonal patterns over Africa, South America, and Asia are now captured more effectively, leading to better alignment with satellite climatologies. Our results confirm that MLS-based partial columns consistently show higher CO values compared to ACE-FTS, with differences largest in the extratropics poleward of 50°. CLaMS simulations align more closely with ACE-FTS than with MLS in that region, and in situ observations in the boreal polar region generally support ACE-FTS rather than MLS. Conversely, in situ measurements agree better with MLS than with ACE-FTS in air affected by the Asian summer monsoon above 10 km.

Sensitivity tests with enhanced boundary-layer CO demonstrate the importance of accurately representing lower boundary conditions for CO and the potential impact of unresolved sources in this region. Incorporating such improvements would further reduce discrepancies with in situ and satellite observations, particularly in convectively active regions. This underscores the need for better spatial and temporal resolution of boundary-layer emissions to enhance model accuracy globally. The improvements in CLaMS-3.0 are not merely incremental; they mark a significant step forward in the model's ability to resolve the intricate transport and mixing processes in the UTLS region. For instance, the enhanced resolution in the PBL and the decoupling of grid adaptation from convection parameterization enable more realistic simulations of convective outflows by applying unresolved convective updrafts every 6 hours and mixing every 24 hours. In future work, we plan to increase the frequency of convection parameterization to 1 hour to further enhance model performance.

In summary, CLaMS-3.0 offers a robust and refined tool for studying UTLS transport processes. The advancements in grid resolution and parameterization of unresolved convection make the model better suited for future studies exploring seasonal and spatial variability in the UTLS region, the impacts of boundary-layer emissions, and the representation of convective transport in a changing climate.

#### 405 Appendix A: Delaunay triangulation



Handling irregular grids, especially determining the nearest neighbors (NNs) in such grids, requires specific mathematical approaches. The most intuitive definition of the NNs for a given point in an irregular grid embedded in a k-dimensional space can be achieved using the concept of the Voronoi volume (or Voronoi area in 2D). The Voronoi volume is defined as the region where every point is closer to the considered point than to any other grid point. In Fig. A1, the light pink-colored region marks the Voronoi area of an arbitrary point in an example 2D grid.

**Figure A1.** Voronoi area (light pink) confined by the Voronoi edges (dashed lines) of an arbitrary grid point within an irregular 2D grid, also defining the volume associated with this point and its nearest neighbors (NNs). Delaunay triangulation (bold lines) is a direct method for finding the NNs.

The borders of the Voronoi regions are called Voronoi edges (dashed lines) and represent the set of points equidistant from two neighboring grid points. In this way, both the volume and the NNs of the considered grid point are defined. Note that Voronoi edges also serve as symmetry axes for pairs of neighboring grid points. Furthermore, every grid point lies within exactly one associated Voronoi region. Two grid points that are nearest neighbors are connected by a line called a Delaunay edge, which forms a network of triangles in 2D, known as the Delaunay triangulation, or in 3D, the Delaunay tetrahedration.

Determining the Delaunay triangulation directly (i.e., without using the concept of Voronoi regions) for an arbitrary irregular grid in k dimensions is a challenging numerical problem. One approach, the convex hull method (Barber et al., 1996), with computational costs scaling as  $N \log N$  (N being the total number of grid points), is based on the idea that finding the Delaunay triangulation for a set of points in k-dimensional Euclidean space can be converted into the problem of finding the convex hull for a set of points in (k+1)-dimensional space (Preparata and Shamos, 1985). The convex hull of a set of points is the smallest convex set containing this set. For a planar set of points, the convex hull can be visualized by imagining an elastic band

stretched around the outermost points; when released, it will assume the shape of the required convex hull. (Note that due to the appropriate projection to a higher dimension, in this example from 1D to 2D, no points will be "left" in the apparent interior of our example 2D set of points).

It can be shown that the concept of layerwise calculation of the NNs in a 3D domain containing N grid points, as used in the CLaMS adaptive grid procedure, is less time-consuming than a full 3D triangulation, which scales as  $N^2$  in the worst case (de Berg et al., 2008). Let us consider m horizontal layers fully filling the 3D domain, with each layer consisting of  $n^2$  grid points. Thus, the aspect ratio  $\alpha = r/\Delta z$ , where r and  $\Delta z$  are the mean horizontal and vertical separation between the grid points, scales as  $\alpha = n/m$ , and the total number of grid points can be rewritten as  $N = mn^2 = n^3/\alpha$ . Consequently, the ratio of the scaling times between the full 3D triangulation,  $t_{3D}$ , and the layer concept,  $t_{2D'}$ , can be written as:

$$\frac{t_{3D}}{t_{2D'}} \sim \frac{N^2}{mn^2 \log n^2} \sim \frac{n^6}{n^3 \log n^2} > \frac{n^6}{n^5} = n \sim N^{\frac{1}{3}}$$
(A1)

where we used the estimate  $\log x < x$ , and n was expressed in terms of N. Thus, as N increases, the 3D triangulation becomes increasingly time-consuming. Furthermore, the 3D triangulation of grid points at the boundaries of a typical atmospheric 3D domain, which extends from the Earth's surface to the upper edge of the atmosphere, is much more complicated compared to the 2D triangulation on closed spherical layers without such boundaries.

## Appendix B: Entropy-static stability grid



Following the procedure described in Konopka et al. (2012), the altitude z-dependent grid parameters r (horizontal separation) and  $\Delta z$  (vertical separation) are given as:

$$r = \left(\frac{kNS_{ap}}{s}\right)^{\frac{1}{3}}, \qquad \Delta z = \left(\frac{1}{kN}\right)^{\frac{2}{3}} \left(\frac{S_{ap}}{s}\right)^{\frac{1}{3}}$$
(B1)

with  $S_{ap}=r_0^2\Delta z_0s_{max}$  denoting the entropy per air parcel with horizontal and vertical extensions  $r_0$  and  $\Delta z_0$ . The 1D profiles of the entropy density s and the dry Brunt-Väisälä frequency  $N_d$  are derived from the US standard atmosphere and are shown in the top and bottom right panels of Fig. B1. The index "0" refers to the altitude of the entropy maximum,  $s_{max}$ , where  $\alpha=\alpha_0=r_0/\Delta z_0=250$  is the expected aspect ratio in the lowermost stratosphere around  $\theta=380$  K (Balluch and Haynes, 1997). The respective value of  $N^2(\theta=380\text{ K})=N_0^2=4.0\cdot10^{-4}\text{ s}^{-2}$  allows us to determine the parameter  $k=\alpha_0/N_0$ . By varying  $r_0$ , i.e., the model horizontal resolution at  $\theta=380$  K (roughly corresponding to the position of the tropical tropopause), different global model resolutions can be chosen, with  $r>r_0$  and  $\Delta z>\Delta z_0$  above and below the level where the 1D entropy s has its maximum (see Fig. 4 in Konopka et al. (2012)).

However, the question arises as to how this idealized 1D approach can be justified by more realistic spatial and temporal distributions of s and N. Although a full 3D approach would be desirable, the climatological all-year zonal means of these two functions, shown in the left part of Fig. B1, may offer some insights into an optimal grid structure.

Following the definition of entropy density,  $s = c_p \rho \ln(\theta/\theta_0)$  (in J/(K·m<sup>3</sup>), with specific heat  $c_p$  (J/(K·kg)) and air mass density  $\rho$  (kg/m<sup>3</sup>), the ERA5-based, all-year 2D zonal mean of s, as a function of latitude and altitude, is shown in the top

**Figure B1.** 2D versus 1D idealized forms of the entropy density s (top) and the dry Brunt-Väisälä frequency  $N_d^2$  (BVF, bottom). The 2D functions represent the ERA5 2005–2020 all-year zonal means (left, black lines are isotachs of zonal winds), while the 1D profiles (right) are derived from the meridionally averaged 2D distributions. For comparison, the 1D profiles based on the US standard atmosphere, as discussed in Konopka et al. (2012), are also shown.

left panel of Fig. B1. The freely chosen reference potential temperature  $\theta_0$  is set to 300 K, as the lowest potential temperature surface that does not cross the Earth's surface. Thus, s values below this level are negative and cannot be used to calculate the grid parameters r and  $\Delta z$  from Eq. (B1). By zonally averaging this 2D distribution, the corresponding 1D profile is shown in the top right panel of Fig. B1, together with the 1D profile derived from the US standard atmosphere (Konopka et al., 2012) (note that for the 1D case  $\theta_0$  is set to the potential temperature at the Earth's surface).


As the 2D s distribution, being a product of the upwardly decreasing air mass density  $\rho$  and the increasing potential temperature  $\theta$ , shows distinct maxima in the polar regions around 15 km and in the tropics around 5 km, the corresponding 1D distribution has a broad maximum around 13-14 km. Consequently, the highest density of APs will be generated in regions different from those suggested by the 2D distribution. Additionally, the 2D s distribution implies that grid generation should be decoupled not only in the PBL but also in the entire region where  $\theta < \theta_0$ . Here,  $\theta \ge \theta_0$  is considered the free atmosphere, and  $\theta < \theta_0$  as an "extended" PBL. Note that the isentropes below  $\theta_0$  cross the Earth's surface and, assuming isentropic motion,

are inherently connected with the PBL. Similar discrepancies in the vertical positioning of maxima can be observed between the 2D and 1D  $N_d^2$  profiles, as shown in the bottom panels of Fig. B1.

Nonetheless, the 1D profiles of s and  $N_d^2$  derived from the US standard atmosphere approximate fairly well those derived from the ERA5 climatology. Additionally, while the seasonality of the 2D distributions may play a role (not shown), it is less pronounced for s than for  $N_d^2$ . Using the 2D distribution to generate the grid would be more physically accurate and desirable, as it could better capture spatial and temporal variability. However, implementing such an approach faces unresolved challenges, such as handling layers that are no longer disjunct, which complicates the grid structure. As a result, the 1D approach remains the practical choice in the current implementation of CLaMS.

## Appendix C: Adaptive grid interpolations

485

Consider two APs with their respective (Voronoi) volumes  $V_1$  and  $V_2$ . The number of molecules  $N_i$  of a trace gas within each AP can be expressed as:

$$N_i = \mu_i n_i V_i, \quad i = 1, 2$$
 (C1)

where  $\mu_i$  and  $n_i$  denote the mixing ratio of the trace gas and the air number density, respectively, for each AP. Assume that these air parcels undergo a mixing event in CLaMS, requiring the mixing ratio of the resulting mixed AP to be derived from the mixing ratios of the original APs.

For the resulting volume V and air density n of the mixed AP, we assume:

$$V = V_1 + V_2$$
 and  $nV = n_1V_1 + n_2V_2$ . (C2)

Using algebra, the mixing ratio  $\mu$  of a trace gas in the mixed AP can be written as:

$$\mu = \frac{N_1 + N_2}{nV} = \frac{\mu_1 n_1 + \eta \mu_2 n_2}{n_1 + \eta n_2} \tag{C3}$$

where  $\eta = V_2/V_1$ . Since both APs lie within the same CLaMS layer, their volumes are approximately equal by construction at initialization. Over time, the adaptive grid procedure may lead to slight inhomogeneities in the AP distribution within a layer due to atmospheric deformation. However, we neglect this effect here and assume  $\eta = 1$  (to approximate  $\eta$ , 2D Voronoi areas, as discussed in Appendix A, can be applied). Note that determining the exact 3D volumes of Lagrangian APs is challenging, making other assumptions for  $\eta$  difficult to implement. After some simplification, eq. (C3) becomes:

$$\mu = \frac{1}{2} \left[ \mu_1(1+\chi) + \mu_2(1-\chi) \right] \tag{C4}$$

with  $\chi = \delta/n$ ,  $n = 0.5 \cdot (n_1 + n_2)$ , and  $\delta = 0.5 \cdot (n_1 - n_2)$ . Using the ideal gas law  $n \sim p/T$ , we derive:

$$\chi = \frac{\delta}{n} = \frac{\frac{p_1}{T_1} - \frac{p_2}{T_2}}{\frac{p_1}{T_1} + \frac{p_2}{T_2}}.$$
 (C5)

## Appendix D: Impact of the MLS Averaging Kernel







Figures 6 and 7 present comparisons of CLaMS and MLS CO partial columns (XCO) calculated over the UTLS range from 147 to 68 hPa for both datasets. While the MLS data are provided at three pressure levels—147, 100, and 68 hPa—with a vertical resolution of approximately 3 km, their true vertical resolution (quantified by the full width at half maximum of the rows of the averaging kernel matrix) is about 5–5.5 km at these altitudes (Livesey et al., 2020). In contrast, the CLaMS data have a much higher vertical resolution, on the order of a few hundred meters. To enable an appropriate comparison between MLS data and CLaMS output, the averaging kernel provided by MLS can be applied to smooth the model data vertically, mimicking the coarser vertical resolution of the satellite. Figure D1 quantifies this effect on both the partial columns and their anomalies.

In the top row of Fig. D1, both quantities are calculated from CLaMS output interpolated onto six pressure levels: 146.8, 121.2, 110.0, 100.0, 90.0, 82.5, 75.0, and 68.0 hPa, which approximately match the vertical resolution of the Lagrangian grid. In the second row, both XCO and its anomaly are calculated on the native MLS CO levels in the UTLS, i.e., 146.8, 100.0, and 68.0 hPa. Additionally, the MLS CO averaging kernel (which is provided on MLS levels) is applied, with the results displayed in the bottom panels of Fig. D1. While the reduction from six levels to three levels increases XCO values by no more than 2.2%, the smoothing with the averaging kernel reduces the column by a maximum of 5%. The effect on the anomalies is even smaller, by an order of magnitude. Consequently, we do not apply the MLS averaging kernel to the CLaMS results in this paper, as its impact is negligible.

## **Appendix E: Partial Columns of Ozone**

To further validate the model's representation of the UTLS region, we complement the CO analysis by evaluating partial columns of ozone (XO<sub>3</sub>), defined as XO<sub>3</sub> =  $\int_{p_{\text{bot}}}^{p_{\text{top}}} (O_3/g) dp$ , where O<sub>3</sub> is the ozone mass mixing ratio, g is the gravitational acceleration, and the integration is carried out between  $p_{\text{bot}} = 147 \text{ hPa}$  and  $p_{\text{top}} = 68 \text{ hPa}$ . This pressure range aligns with native Microwave Limb Sounder (MLS) levels and thus facilitates a direct comparison with satellite data.

XO<sub>3</sub> values are derived from CLaMS-3.0/1.0 simulations and compared with MLS observations. As with CO, we find that the integral values are only weakly affected by whether the MLS averaging kernel is applied to the model output (difference below 10%, not shown).

While Fig. E1 shows the seasonality of the absolute values, the anomalies are shown in Fig. E2. The correlation coefficients between XO<sub>3</sub> absolute values from CLaMS-3.0 and MLS range from 0.82 (SON) to 0.99 (JJA), with a value of 0.95 for the multi-seasonal average (ALL), and are even higher for the anomalies. While the overall improvement compared to CLaMS-1.0 is minor in terms of correlation, the difference patterns, CLaMS-1.0 minus CLaMS-3.0 (last column in Fig. E1), show a clearer signal in the tropics. In particular, over the Maritime Continent, CLaMS-3.0 captures stronger lofting of ozone-poor marine boundary layer air into the UTLS, consistent with the enhanced role of convection and in line with the CO-based results discussed in Sect. 4.

These findings confirm that the model developments presented here primarily influence the tropospheric part of the UTLS, where CO — rather than ozone — is a more suitable tracer. Since the model's lower ozone boundary (approximating the

**Figure D1.** Annual mean climatologies of the CO partial column (XCO, in mg/m<sup>2</sup>) for the period 2005–2020, spanning from 147 to 68 hPa (left) and their respective zonal anomalies ( $\Delta$ XCO, right). Results are shown from CLaMS interpolated to six pressure levels (top), to three MLS pressure levels (middle), and with the MLS averaging kernel applied to these three levels (bottom).

**Figure E1.** Seasonal (DJF, MAM, JJA, and SON) values of the O<sub>3</sub> partial column, XO<sub>3</sub>, derived from the 2005–2020 climatology for MLS (left), CLaMS-3.0 (middle), and CLaMS-1.0 (right; shown as the percentage difference relative to CLaMS-3.0).

Figure E2. Same as Fig. E1, but for  $O_3$  partial column seasonal anomalies,  $\Delta XO_3$ , calculated for CLaMS-3.0 relative to the zonal mean for each respective season.

PBL at 1.6–2.2 km) is set to zero and tropospheric ozone chemistry is neglected (Pommrich et al., 2014), we refrain from overinterpreting the absolute XO<sub>3</sub> values. Nevertheless, as a stratospheric tracer, ozone remains well represented, with CLaMS-3.0 retaining the key features already captured by CLaMS-1.0 (Konopka et al., 2010; Yan et al., 2018), while extending model improvements into the lower UTLS.

Acknowledgements. The authors gratefully acknowledge the European Centre for Medium-Range Weather Forecasts (ECMWF) for providing meteorological analysis for this study. The Atmospheric Chemistry Experiment (ACE), also known as SCISAT, is a Canadian-led mission mainly supported by the Canadian Space Agency and the Natural Sciences and Engineering Research Council of Canada. Excellent programming support was provided by Nicole Thomas. Discussions with Jens-Uwe Grooß also motivated some of the tropospheric extensions in CLaMS-3.0. Funding for this work was provided by the Earth System Modelling Project (ESM) through the CLaMS-ESM project, which also provided computing time on the ESM partition of the supercomputer JUWELS at the Jülich Supercomputing Centre (JSC). Work at the Jet Propulsion Laboratory, California Institute of Technology, was carried out under a contract with the National Aeronautics and Space Administration (80NM0018D0004). This work is published under the Creative Commons Attribution 4.0 License, supporting open access and unrestricted sharing of research findings. Felix Ploeger has been supported by the Deutsche Forschungsgemeinschaft (DFG, German Research Foundation; TRR 301, project ID 428312742).

Code and data availability. CLaMS-3.0/MESSy is available as part of the Modular Earth Submodel System (MESSy), Version 2.54.0, accessible at the Mercurial server: http://messy.fz-juelich.de/messy-2.54.0-clams. ERA5 reanalysis data are provided by the European Centre for Medium-Range Weather Forecasts (ECMWF) and are available as deterministic forecasts (atmospheric model) via https://apps.ecmwf. int/data-catalogues/era5/?class=ea. ERA-Interim data are also available via https://apps.ecmwf.int/archive-catalogue/?class=ei. MLS data used in this study are from Version 5.1 and are accessible from the NASA Goddard Earth Sciences Data and Information Services Center (GES DISC) at https://disc.gsfc.nasa.gov/datasets?page=1&keywords=MLS%20Level%202. The ACE-FTS data used in this study are from Version 5.2 and are available from the ACE mission website at https://ace.scisat.ca, with access details provided in Boone et al. (2023); Sheese and Walker (2023). The RECONCILE in situ measurement data are available via the HALO database at https://halo-db.pa.op.dlr.de/mission/7. The WISE campaign data are available from the HALO database at https://halo-db.pa.op.dlr.de/mission/11. The ACCLIP campaign data are hosted by the NCAR Earth Observing Laboratory (EOL) and can be accessed at https://data.eol.ucar.edu/master\_list/?project=ACCLIP. The PHILEAS campaign data will be available through the HALO database at https://halo-db.pa.op.dlr.de/mission/15 (data access pending public release).

- ACCLIP Data Reference, https://doi.org/10.26023/2F7P-XVDS-S60Y, 2023.
- Abalos, M., Randel, W. J., Kinnison, D., and Garcia, R.: Using the artificial tracer e90 to examine present and future UTLS tracer transport in WACCM, J. Geophys. Res., 74, 3383–3403, https://doi.org/10.1029/2002JD002634, 2017.
- Balluch, M. G. and Haynes, P. H.: Quantification of lower stratospheric mixing processes using aircraft data, J. Geophys. Res., 102, 23 487–555 23 504, 1997.
  - Barber, C. B., David, P. D., and Huhdanpaa, H.: The Quickhull Algorithm for Convex Hulls, ACM Transactions on Mathematical Software, 22, 469–483, 1996.
  - Boone, C. D., Bernath, P. F., and Lecours, M.: Version 5 retrievals for ACE-FTS and ACE-imagers, Journal of Quantitative Spectroscopy and Radiative Transfer, 310, 108749, https://doi.org/10.1016/j.jqsrt.2023.108749, 2023.
- Charlesworth, E., Ploeger, F., Birner, T., Baikhadzhaev, R., Abalos, M., Abraham, N. L., Akiyoshi, H., Bekki, S., Dennison, F., Jöckel, P., Keeble, J., Kinnison, D., Morgenstern, O., Plummer, D., Rozanov, E., Strode, S., Zeng, G., Egorova, T., and Riese, M.: Stratospheric water vapor affecting atmospheric circulation, Nature Communications, 14, 3925, https://doi.org/10.1038/s41467-023-39559-2, 2023.
  - Courant, R., Friedrichs, K., and Lewy, H.: Über die partiellen Differenzengleichungen der mathematischen Physik, Mathematische Annalen, 100, 32–74, 1928.
- de Berg, M., Cheong, O., van Kreveld, M., and Overmars, M.: Computational Geometry: Algorithms and Applications, Springer-Verlag, 3rd edn., ISBN 978-3-540-77973-5, 2008.
  - Grise, K. M., Thompson, D. W. J., and Birner, T.: A Global Survey of Static Stability in the Stratosphere and Upper Troposphere, Journal of Climate, 23, 2275–2292, https://doi.org/10.1175/2009JCLI3369.1, 2010.
- Haynes, P. and Anglade, J.: The vertical scale cascade in atmospheric tracers due to large-scale differential advection, J. Atmos. Sci., 54, 1121–1136, 1997.
  - Hegglin, M. I., Boone, C. D., Manney, G. L., and Walker, K. A.: A global view of the extratropical tropopause transition layer from Atmospheric Chemistry Experiment Fourier Transform Spectrometer O3, H2O, and CO, Journal of Geophysical Research: Atmospheres, 114, https://doi.org/10.1029/2008JD009984, 2009.
- Hegglin, M. I., Tegtmeier, S., Anderson, J., Funke, B., Gille, J., Kellmann, S., Kinnison, D., Kyrölä, E., Lumpe, J., and Walker, K. A.:

  Overview and update of the SPARC Data Initiative: comparison of stratospheric composition measurements from satellite limb sounders,
  Earth System Science Data, 13, 1855–1903, https://doi.org/10.5194/essd-13-1855-2021, 2021.
  - Hersbach, H., Bell, B., Berrisford, P., Hirahara, S., Horányi, A., Muñoz-Sabater, J., Nicolas, J., Peubey, C., Radu, R., Schepers, D., Simmons, A., Soci, C., Abdalla, S., Abellan, X., Balsamo, G., Bechtold, P., Biavati, G., Bidlot, J., Bonavita, M., De Chiara, G., Dahlgren, P., Dee, D., Diamantakis, M., Dragani, R., Flemming, J., Forbes, R., Fuentes, M., Geer, A., Haimberger, L., Healy, S., Hogan, R. J., Hólm, E.,
- Janisková, M., Keeley, S., Laloyaux, P., Lopez, P., Lupu, C., Radnoti, G., de Rosnay, P., Rozum, I., Vamborg, F., Villaume, S., and Thépaut, J.-N.: The ERA5 global reanalysis, Q. J. R. Meteorol. Soc., 146, 1999–2049, https://doi.org/10.1002/qj.3803, 2020.
  - Jöckel, P., Kerkweg, A., Pozzer, A., Sander, R., Tost, H., Riede, H., Baumgaertner, A., Gromov, S., and Kern, B.: Development cycle 2 of the Modular Earth Submodel System (MESSy2), Geoscientific Model Development, 3, 717–752, https://doi.org/10.5194/gmd-3-717-2010, 2010.
- Jülich Supercomputing Centre: JUWELS: Modular Tier-0/1 Supercomputer at the Jülich Supercomputing Centre, Journal of large-scale research facilities, 5, A135, http://dx.doi.org/10.17815/jlsrf-5-171, 2019.

- Kloss, C., von Hobe, M., Höpfner, M., Walker, K. A., Riese, M., Ungermann, J., Hassler, B., Kremser, S., and Bodeker, G. E.: Sampling bias adjustment for sparsely sampled satellite measurements applied to ACE-FTS carbonyl sulfide observations, Atmospheric Measurement Techniques, 12, 2129–2138, https://doi.org/10.5194/amt-12-2129-2019, 2019.
- Konopka, P., Steinhorst, H.-M., Grooß, J.-U., Günther, G., Müller, R., Elkins, J. W., Jost, H.-J., Richard, E., Schmidt, U., Toon, G., and McKenna, D. S.: Mixing and Ozone Loss in the 1999-2000 Arctic Vortex: Simulations with the 3-dimensional Chemical Lagrangian Model of the Stratosphere (CLaMS), J. Geophys. Res., 109, D02315, https://doi.org/10.1029/2003JD003792, 2004.

595

- Konopka, P., Grooß, J.-U., Günther, G., Ploeger, F., Pommrich, R., Müller, R., and Livesey, N.: Annual cycle of ozone at and above the tropical tropopause: observations versus simulations with the Chemical Lagrangian Model of the Stratosphere (CLaMS), Atmospheric Chemistry and Physics, 10, 121–132, https://doi.org/10.5194/acp-10-121-2010, 2010.
- Konopka, P., Ploeger, F., and Müller, R.: Entropy- and static stability-based Lagrangian model grids, in: Geophysical Monograph Series: Lagrangian Modeling of the Atmosphere, edited by Lin, J., vol. 200, pp. 99–109, American Geophysical Union, https://doi.org/10.1029/2012GM001253, 2012.
- Konopka, P., Tao, M., Ploeger, F., Diallo, M., and Riese, M.: Tropospheric mixing and parametrization of unresolved convective updrafts as implemented in the Chemical Lagrangian Model of the Stratosphere (CLaMS v2.0), Geosci. Model Dev., 12, 2441–2462, https://doi.org/doi:10.5194/gmd-12-2441-2019, 2019.
  - Konopka, P., Tao, M., von Hobe, M., Hoffmann, L., Kloss, C., Ravegnani, F., Volk, C. M., Lauther, V., Zahn, A., Hoor, P., and Ploeger, F.: Tropospheric transport and unresolved convection: numerical experiments with CLaMS 2.0/MESSy, Geoscientific Model Development, 15, 7471–7487, https://doi.org/10.5194/gmd-15-7471-2022, 2022.
- Krasauskas, L., Ungermann, J., Preusse, P., Friedl-Vallon, F., Zahn, A., Ziereis, H., Rolf, C., Plöger, F., Konopka, P., Vogel, B., and Riese, M.: 3-D tomographic observations of Rossby wave breaking over the North Atlantic during the WISE aircraft campaign in 2017, Atmos. Chem. Phys., 21, 10249–10272, https://doi.org/10.5194/acp-21-10249-2021, 2021.
  - Krüger, K., Tegtmeier, S., and Rex, M.: Long-term climatology of air mass transport through the Tropical Tropopause Layer (TTL) during NH winter, Atmospheric Chemistry and Physics, 8, 813–823, https://doi.org/10.5194/acp-8-813-2008, 2008.
- Lauther, V., Vogel, B., Wintel, J., Rau, A., Hoor, P., Bense, V., Müller, R., and Volk, C. M.: In situ observations of CH<sub>2</sub>Cl<sub>2</sub> and CHCl<sub>3</sub> show efficient transport pathways for very short-lived species into the lower stratosphere via the Asian and North American summer monsoons, Atmos. Chem. Phys., 2021, 1–42, https://doi.org/10.5194/acp-2021-837, 2021.
  - Legras, B. and Bucci, S.: Confinement of air in the Asian monsoon anticyclone and pathways of convective air to the stratosphere during the summer season, Atmospheric Chemistry and Physics, 20, 11 045–11 064, https://doi.org/10.5194/acp-20-11045-2020, 2020.
- Livesey, N. J., Read, W. G., Wagner, P. A., Froidevaux, L., Santee, M. L., Schwartz, M. J., Lambert, A., Millan Valle, L. F., Pumphrey, H. C., Manney, G. L., Fuller, R. A., Jarnot, R. F., Knosp, B. W., and Lay, R. R.: Version 5.0x Level 2 and 3 data quality and description document, Tech. Rep. JPL D-105336 Rev. A, Jet Propulsion Laboratory, California Institute of Technology Pasadena, California, 91109-8099, http://mls.jpl.nasa.gov, 2020.
- Mahowald, N. M., Plumb, R. A., Rasch, P. J., del Corral, J., and Sassi, F.: Stratospheric transport in a three-dimensional isentropic coordinate model, J. Geophys. Res., 107, 4254, https://doi.org/10.1029/2001JD001313, 2002.
  - McKenna, D. S., Grooß, J.-U., Günther, G., Konopka, P., Müller, R., Carver, G., and Sasano, Y.: A new Chemical Lagrangian Model of the Stratosphere (CLaMS): 2. Formulation of chemistry scheme and initialization, J. Geophys. Res., 107, 4256, https://doi.org/10.1029/2000JD000113, 2002a.

- McKenna, D. S., Konopka, P., Grooß, J.-U., Günther, G., Müller, R., Spang, R., Offermann, D., and Orsolini, Y.: A new Chemical Lagrangian Model of the Stratosphere (CLaMS): 1. Formulation of advection and mixing, J. Geophys. Res., 107, 4309, https://doi.org/10.1029/2000JD000114, 2002b.
  - McManus, J. B., Zahniser, M. S., Nelson, J. D. D., Shorter, J. H., Herndon, S., Wood, E., and Wehr, R.: Application of quantum cascade lasers to high-precision atmospheric trace gas measurements, Optical Engineering, 49, 111 124–1-111 124–11, 2010.
- Pan, L. L., Konopka, P., and Browell, E. V.: Observations and model simulations of mixing near the extratropical tropopause, J. Geophys. Res., 111, D05106, https://doi.org/10.1029/2005JD006480, 2006.
  - Pan, L. L., Atlas, E. L., Honomichl, S. B., Smith, W. P., Kinnison, D. E., Solomon, S., Santee, M. L., Saiz-Lopez, A., Laube, J. C., Wang, B., Ueyama, R., Bresch, J. F., Hornbrook, R. S., Apel, E. C., Hills, A. J., Treadaway, V., Smith, K., Schauffler, S., Donnelly, S., Hendershot, R., Lueb, R., Campos, T., Viciani, S., D'Amato, F., Bianchini, G., Barucci, M., Podolske, J. R., Iraci, L. T., Gurganus, C., Bui, P., Dean-Day, J. M., Millán, L., Ryoo, J.-M., Barletta, B., Koo, J.-H., Kim, J., Liang, Q., Randel, W. J., Thornberry, T., and Newman, P. A.: East Asian summer monsoon delivers large abundances of very short-lived organic chlorine substances to the lower stratosphere, Proceedings of the
  - Park, M., Randel, W. J., Gettelman, A., Massie, S. T., and Jiang, J. H.: Transport pathways of carbon monoxide in the Asian summer monsoon diagnosed from Model of Ozone and Related Tracers (MOZART), Journal of Geophysical Research: Atmospheres, 114, D08 303, https://doi.org/10.1029/2008JD010621, 2009.
- Ploeger, F., Konopka, P., Günther, G., Grooß, J.-U., and Müller, R.: Impact of the vertical velocity scheme on modeling transport across the tropical tropopause layer, J. Geophys. Res., 115, D03301, https://doi.org/10.1029/2009JD012023, 2010.

National Academy of Sciences, 121, e2318716121, https://doi.org/10.1073/pnas.2318716121, 2024.

- Ploeger, F., Birner, T., Charlesworth, E., Konopka, P., and Müller, R.: Moist bias in the Pacific upper troposphere and lower stratosphere (UTLS) in climate models affects regional circulation patterns, Atmospheric Chemistry and Physics, 24, 2033–2043, https://doi.org/10.5194/acp-24-2033-2024, 2024.
- Pommrich, R., Müller, R., Grooß, J.-U., Konopka, P., Ploeger, F., Vogel, B., Tao, M., Hoppe, C. M., Günther, G., Spelten, N., Hoffmann, L., Pumphrey, H.-C., Viciani, S., D'Amato, F., Volk, C. M., Hoor, P., Schlager, H., and Riese, M.: Tropical troposphere to stratosphere transport of carbon monoxide and long-lived trace species in the Chemical Lagrangian Model of the Stratosphere (CLaMS), Geoscientific Model Development, 7, 2895–2916, https://doi.org/10.5194/gmd-7-2895-2014, 2014.
- Poujol, B. and Bony, S.: Measuring Clear-Air Vertical Motions From Space, AGU Advances, 5, e2024AV001267, https://doi.org/https://doi.org/10.1029/2024AV001267, e2024AV001267 2024AV001267, 2024.
  - Prather, M., Zhu, X., Tang, Q., Hsu, J., and Neu, J.: An atmospheric chemist in search of the tropopause, J. Geophys. Res., 116, 2011.
  - Preparata, F. P. and Shamos, M.: Computational Geometry. An Introduction, Springer-Verlag, 1985.

635

- Press, W. H., Teukolsky, S. A., Vetterling, W. T., and Flannery, B. P.: Numerical Recipes: The Art of Scientific Computing, Cambridge University Press, Cambridge, UK, 3rd edn., ISBN 978-0521880688, 2007.
- Pugh, T. A. M., Cain, M., Methven, J., Wild, O., Arnold, S. R., Real, E., Law, K. S., Emmerson, K. M., Owen, S. M., Pyle, J. A., Hewitt, C. N., and MacKenzie, A. R.: A Lagrangian model of air-mass photochemistry and mixing using a trajectory ensemble: the Cambridge Tropospheric Trajectory model of Chemistry And Transport (CiTTyCAT) version 4.2, Geosci. Model Dev., 5, 193–221, 2012.
  - Reithmeier, C. and Sausen, R.: ATTILA Atmospheric Tracer Transport in a Lagrangian Model, Tech. rep., Institut für Physik der Atmosphäre, DLR-Oberpfaffenhofen, 82234 Wessling, Germany, 2000.

- Santee, M. L., Manney, G. L., Livesey, N. J., Schwartz, M. J., Neu, J. L., and Read, W. G.: A comprehensive overview of the climatological composition of the Asian summer monsoon anticyclone based on 10 years of Aura Microwave Limb Sounder measurements, Journal of Geophysical Research: Atmospheres, 122, 5491–5514, https://doi.org/10.1002/2016JD026408, 2017.
  - Schoeberl, M. R. and Dessler, A. E.: Dehydration of the stratosphere, Atmospheric Chemistry and Physics, 11, 8433–8446, https://doi.org/10.5194/acp-11-8433-2011, 2011.
- 665 Shannon, C. E.: A Mathematical Theory of Communication, The Bell System Technical Journal, 27, 379–423, 623–656, https://doi.org/10.1002/j.1538-7305.1948.tb01338.x, 1948.
  - Sheese, P. and Walker, K.: Data Quality Flags for ACE-FTS Level 2 Version 5.2 Data Set, https://doi.org/10.5683/SP3/NAYNFE, 2023.
  - Sheese, P. E., Boone, C. D., and Walker, K. A.: Detecting physically unrealistic outliers in ACE-FTS atmospheric measurements, Atmospheric Measurement Techniques, 8, 741–750, https://doi.org/10.5194/amt-8-741-2015, 2015.
- 670 Stenke, A., Grewe, V., and Ponater, M.: Lagrangian transport of water vapor and cloud water in the ECHAM4 GCM and its impact on the cold bias, Climate Dynamics, 31, 491–506, https://doi.org/10.1007/s00382-007-0347-5, 2008.
  - Stenke, A., Dameris, M., Grewe, V., and Garny, H.: Implications of Lagrangian transport for simulations with a coupled chemistry-climate model, Atmospheric Chemistry and Physics, 9, 5489–5504, https://doi.org/10.5194/acp-9-5489-2009, 2009.
- Tao, M., Pan, L. L., Konopka, P., Honomichl, S. B., Kinnison, D. E., and Apel, E. C.: A Lagrangian Model Diagnosis of

  Stratospheric Contributions to Tropical Midtropospheric Air, Journal of Geophysical Research: Atmospheres, 123, 9764–9785,

  https://doi.org/https://doi.org/10.1029/2018JD028696, 2018.
  - Viciani, S., D'Amato, F., Mazzinghi, P., Castagnoli, F., Toci, G., and Werle, P.: A cryogenically operated laser diode spectrometer for airborne measurement of stratospheric trace gases, Applied Physics B, 90, 581–592, https://doi.org/10.1007/s00340-007-2885-2, 2008.
- Viciani, S., Montori, A., Chiarugi, A., and D'Amato, F.: A Portable Quantum Cascade Laser Spectrometer for Atmospheric Measurements of Carbon Monoxide, Sensors, 18, 2380, https://doi.org/10.3390/s18072380, 2018.
  - Vogel, B., Günther, G., Müller, R., Grooß, J.-U., Afchine, A., Bozem, H., Hoor, P., Krämer, M., Müller, S., Riese, M., Rolf, C., Spelten, N., Stiller, G. P., Ungermann, J., and Zahn, A.: Long-range transport pathways of tropospheric source gases originating in Asia into the northern lower stratosphere during the Asian monsoon season 2012, Atmospheric Chemistry and Physics, 16, 15 301–15 325, https://doi.org/10.5194/acp-16-15301-2016, 2016.
- von Hobe, M., Bekki, S., Borrmann, S., Cairo, F., D'Amato, F., Di Donfrancesco, G., Dörnbrack, A., Ebersoldt, A., Ebert, M., Emde, C., Engel, I., Ern, M., Frey, W., Genco, S., Griessbach, S., Grooß, J.-U., Gulde, T., Günther, G., Hösen, E., Hoffmann, L., Homonnai, V., Hoyle, C. R., Isaksen, I. S. A., Jackson, D. R., Jánosi, I. M., Jones, R. L., Kandler, K., Kalicinsky, C., Keil, A., Khaykin, S. M., Khosrawi, F., Kivi, R., Kuttippurath, J., Laube, J. C., Lefèvre, F., Lehmann, R., Ludmann, S., Luo, B. P., Marchand, M., Meyer, J., Mitev, V., Molleker, S., Müller, R., Oelhaf, H., Olschewski, F., Orsolini, Y., Peter, T., Pfeilsticker, K., Piesch, C., Pitts, M. C., Poole, L. R., Pope,
- F. D., Ravegnani, F., Rex, M., Riese, M., Röckmann, T., Rognerud, B., Roiger, A., Rolf, C., Santee, M. L., Scheibe, M., Schiller, C., Schlager, H., Siciliani de Cumis, M., Sitnikov, N., Søvde, O. A., Spang, R., Spelten, N., Stordal, F., Sumińska-Ebersoldt, O., Ulanovski, A., Ungermann, J., Viciani, S., Volk, C. M., vom Scheidt, M., von der Gathen, P., Walker, K., Wegner, T., Weigel, R., Weinbruch, S., Wetzel, G., Wienhold, F. G., Wohltmann, I., Woiwode, W., Young, I. A. K., Yushkov, V., Zobrist, B., and Stroh, F.: Reconciliation of essential process parameters for an enhanced predictability of Arctic stratospheric ozone loss and its climate interactions (RECONCILE):
   activities and results, Atmospheric Chemistry and Physics, 13, 9233–9268, https://doi.org/10.5194/acp-13-9233-2013, 2013.
  - von Hobe, M., Ploeger, F., Konopka, P., Kloss, C., Ulanowski, A., Yushkov, V., Ravegnani, F., Volk, C. M., Pan, L. L., Honomichl, S. B., Tilmes, S., Kinnison, D. E., Garcia, R. R., and Wright, J. S.: Upward transport into and within the Asian monsoon anticyclone as inferred

- from StratoClim trace gas observations, Atmospheric Chemistry and Physics, 21, 1267–1285, https://doi.org/10.5194/acp-21-1267-2021, 2021.
- Wohltmann, I. and Rex, M.: The Lagrangian chemistry and transport model ATLAS: validation of advective transport and mixing, Geosci. Model Dev., 2, 153–173, https://doi.org/10.5194/gmd-2-153-2009, 2009.
  - Wright, J. S., Zhang, S., Chen, J., Davis, S. M., Konopka, P., Lu, M., Yan, X., and Zhang, G. J.: Evaluating reanalysis representations of climatological trace gas distributions in the Asian monsoon tropopause layer, EGUsphere, 2025, 1–36, https://doi.org/10.5194/egusphere-2025-135, 2025.
- Yan, X., Konopka, P., Ploeger, F., Tao, M., Müller, R., Santee, M. L., Bian, J., and Riese, M.: El Niño Southern Oscillation influence on the Asian summer monsoon anticyclone, Atmospheric Chemistry and Physics, 18, 8079–8096, https://doi.org/10.5194/acp-18-8079-2018, 2018.