# Peer review of "Isentropic Mixing vs. Convection in CLaMS-3.0/MESSy: Evaluation Using Satellite Climatologies and In Situ Carbon Monoxide Observations"

_EGUsphere, 2025_

## Author Response (AR1)

**Response to Referee 1**

We thank Referee 1 for the thoughtful and detailed feedback. We appreciate all comments which clearly helped to improve the manuscript, and we addressed all points in the revised version. Reviewer comments are in black, answers in green.

**General assessment**

*Reviewer:* This is a very well written paper which is easy to understand and I recommend it be accepted after very minor corrections noted below.

**Author response:** We thank the reviewer for this positive assessment. We have implemented the suggested minor corrections and clarified several points as detailed below.

**Hybrid grid, gravity waves, and $q_e$**

*Reviewer:* The hybrid grid used in the tropospheric portion of CLaMS blends kinematic and isentropic trajectory schemes to get around the conservation problems with potential temperature ($\theta$) in the troposphere and surface issues. The advantage of the isentropic schemes is that they are immune to gravity wave biases whereas kinematic schemes can suffer from these biases. For example, the vertical motion of a partially resolved gravity wave or convective complex can dominate the smaller motions associated with larger scale waves. This issue is discussed in the text, but the authors should also discuss how the model would work for orographic gravity waves. For convection, an alternate approach to using a hybrid coordinate is to use $q_e$ which automatically includes the latent heat energy. Convective parcels rise roughly to the $\theta$ level that is equivalent to $q_e$ at the surface. I was surprised that this idea wasn't mentioned.

**Author response:** Yes, resolved gravity waves can still be a problem. The performance of the hybrid coordinate $\zeta$ can be slightly improved by increasing the parameter $\sigma_r$, making the coordinate more diabatic in regions affected by orography. This point was discussed in Appendix A of Tao et al. (2018). Their Fig. A1 (included below) illustrates the advantage of choosing $\sigma_r = 0.7$ over $\sigma_r = 0.3$, as the adiabatic coordinate surfaces then extend deeper into the troposphere compared to our standard configuration.
Concerning the use of $q_e$, we completely agree with the reviewer's idea. One problem is that the quality of water vapor in the reanalysis, which is used in the definition of $q_e$, is not good enough to guarantee a physically useful coordinate. A reliable humidity dataset would be needed throughout the atmosphere (e.g. from meteorological analysis or reanalysis). However,

[Figure]

Figure 1: Orographic gravity waves can be better represented using a more diabatic hybrid vertical coordinate $\zeta$ with $\sigma_r = 0.7$ (i.e., $p_r = 700$ hPa). Shown are dry isentropes (thick dashed lines) and $\zeta$-surfaces (solid black lines).

in ERA5 (and similarly in other (re-)analyses) there are even missing values — especially in regions affected by convection — which makes the use of ERA5 specific humidity problematic. However, this approach is certainly worth revisiting in the context of ERA6 data. Following this recommendation we changed some text, see lines 181-190 in the revised version.

**Mixing scheme and computational cost**

*Reviewer:* CLaMS, unlike most trajectory models, employs a mixing scheme where nearby parcels are 'mixed' as might occur in the actual atmosphere. The scheme also includes creation of new parcels as part of an entropy conserving regridding. The authors note that the regridding step is computationally costly since it involves identifying and estimating distance to the nearest neighbors. I have implemented regridding schemes using the Lyapunov approach (similar to the authors) and since my code was highly parallelized, the computational cost of regridding was larger than the cost just accumulating parcels. However, the cost of accumulating parcels for multi-year runs is also cost prohibitive. The authors do a good job describing their approach and are honest about the computational cost which can be reduced by adjusting the frequency of regridding.

**Author response:** Thank you for the constructive comment — we fully agree. In our experience as well, the accumulation of parcels in long Lagrangian simulations without regridding becomes a major limiting factor.

**Discussion of the three CLaMS versions**

*Reviewer:* The discussion of the three versions of CLaMS was very interesting and enlightening. Clearly CLaMS 3.0 is a step forward from early versions (Fig. 4, 5).

**Author response:** Thanks for your positive judgement.

**CO simulations and MLS/ACE-FTS disagreement**

*Reviewer:* The test with CO simulations is also very interesting and a little confounding in that MLS and ACE-FTS don't agree on the partial columns. The CLaMS simulation looks good when boundary CO is increased (an interesting result in itself). Comparison with in situ data is also useful showing the improvements in 3.0.

**Author response:** Thank you. We agree that the discrepancy between MLS and ACE-FTS CO partial columns is indeed puzzling. In this study, we acknowledge this disagreement but focus our discussion on the model simulations, which show a clear improvement in CO representation with CLaMS-3.0 compared to earlier versions.

**Specific minor suggestions**

1. *Reviewer:* Comment on how CLaMS hybrid coordinate deals with orographic gravity waves. This is probably the worst-case scenario for hybrid coordinates. I suggest an experiment where hybrid and isentropic calculations are compared near mountain ranges.

   **Author response:** See our response above under "Hybrid grid, gravity waves, and $q_e$".

2. *Reviewer:* Add black dots to the caption in Fig. 8.

   **Author response:** Done.

**References**

Tao, M., Pan, L. L., Konopka, P., Honomichl, S. B., Kinnison, D. E., and Apel, E. C. (2018). A lagrangian model diagnosis of stratospheric contributions to tropical midtropospheric air. *Journal of Geophysical Research: Atmospheres*, 123(17):9764–9785.

**Response to Referee 2**

We thank Referee 2 for the constructive and careful review. We appreciate the thoughtful criticism regarding scope, novelty, and evaluation strategy. Reviewer comments are in black, answers in green.

**General assessment**

*Reviewer:* This manuscript presents an update of the Chemical Lagrangian Model of the Stratosphere (CLaMS), focusing on improvements in the parameterization of convection and the adaptive grid procedure used in CLaMS-3.0/MESSy. The authors evaluate the impact of these modifications on the simulation of CO partial columns in the UTLS using satellite observations (MLS, ACE-FTS), as well as CO profiles in the troposphere and UTLS using various airborne campaigns.

While the paper is well written and technically sound, I have reservations regarding its suitability for Atmospheric Chemistry and Physics. My concerns are twofold: (1) the lack of scientific novelty beyond incremental model development; and (2) the scope of the study, which limits its relevance to a broader atmospheric chemistry or physics audience, making it primarily of interest to users and developers of the CLaMS model. The manuscript presents valuable technical developments that merit publication, but it would be more appropriately considered for Geoscientific Model Development (GMD), where the scope and format better match the nature of this work.

**Author response:** We thank the reviewer for this overall positive general assessment. For a paper focussing on both scientific and technical aspects, like the present one, it is often a subtle question if ACP or GMD would be the more appropriate journal. We discussed this question among the authors and, based on the balance between scientific and technical content, came to the conclusion that ACP would be the better choice. Also the other reviewer does not argue against that decision. For more detailed answers regarding the reviewer's concern about the suitability of this paper for ACP, we kindly refer to our response to Major Comment 2.

**Major Comment 1: Convection parameterization builds on earlier work**

*Reviewer:* The convection parameterization implemented here builds on earlier work (Konopka et al., 2012, 2019, 2022). The improvements presented in this paper include the use of CAPE as a convective trigger, increased resolution in the PBL, and the decoupling of the convective time

step (6 h) from the mixing step (24 h). These are technical refinements rather than conceptual breakthroughs.

**Author response:** We agree that the convection parameterization in CLaMS-3.0 builds on previous work (Konopka et al., 2019, 2022). However, convection remains a particularly challenging process to represent in atmospheric models, especially within the Lagrangian framework based on a diabatic transport approach. In our opinion, replacing the wet Brunt–Väisälä frequency with Convective Available Potential Energy (CAPE) as the convective trigger is not just a technical refinement but also a physically more justified choice.

Furthermore, the decoupling of the convective time step (6 h) from the mixing step (24 h) reflects the physical distinction between fast convective events and slower, synoptic-scale mixing processes. Finally, the increased spatial resolution in the Planetary Boundary Layer (PBL), guided by the Shannon entropy concept, is likewise motivated by physical reasoning rather than implementation constraints. We therefore view these changes as meaningful improvements to the physical realism of the model.

**Major Comment 2: Limited novelty / model engineering updates**

*Reviewer:* While they do lead to improved agreement with satellite and in situ data compared to CLaMS v1, these improvements do not lead to new insights into atmospheric chemistry or physics, and are best characterized as model engineering updates. In my view, they fall more within the scope of Geoscientific Model Development (GMD).

**Author response:** We fully agree that model formulation and documentation alone would not justify publication in ACP. Our motivation for submitting this paper to ACP rests on two main points:

(1) The paper goes beyond model updates by providing a detailed and systematic evaluation of convective transport from the Planetary Boundary Layer (PBL) to the Upper Troposphere and Lower Stratosphere (UTLS), using carbon monoxide (CO) as a tracer. This includes a climatological comparison between Microwave Limb Sounder (MLS) and ACE-FTS satellite products, as well as in situ observations from recent aircraft campaigns. These comparisons not only reveal strengths and limitations of each dataset but also uncover a significant discrepancy in CO partial columns between MLS and ACE-FTS. The associated model-based analysis helps to explain this difference and sheds light on the underlying transport processes — insights relevant to the broader atmospheric community.

(2) Our study promotes the use of CO as an ideal tracer for evaluating transport in models, as

exemplified by the validation of CLaMS-3.0. CO is particularly well suited due to its intermediate atmospheric lifetime and the broad availability of satellite (MLS, ACE-FTS, AIRS) and in situ measurements. Even the lower boundary condition for CO, approximating surface concentrations, can be derived from satellite observations (e.g., AIRS). This makes CO a powerful diagnostic for validating model representations of convective transport from the PBL to the UTLS across multiple spatial and temporal scales (of about 100 days).

**Major Comment 3: Consider evaluating other tracers (e.g., ozone)**

*Reviewer:* The authors use CO as a passive tracer of transport from the PBL to the UTLS. I would encourage the authors to consider evaluating other tracers such as ozone, particularly in regions where convective transport plays a dominant role—e.g., the upper troposphere over the Maritime Continent, where ozone minima are observed due to the lofting of ozone-poor marine boundary layer air into the UTLS.

   **Author response:** We thank the reviewer for this excellent suggestion. In the revised manuscript, we have added ozone as a complementary tracer to CO; the results are presented in the new Appendix E. As the reviewer correctly anticipated, CLaMS-3.0 shows an improved representation of convective transport, including more pronounced ozone minima over the Maritime Continent due to enhanced lofting of ozone-poor marine boundary layer air into the UTLS. However, since ozone in CLaMS-3.0 is still not sufficiently well represented in the troposphere (no detailed tropospheric chemistry, ozone is set to zero in the PBL), we refrain from drawing quantitative conclusions. Nevertheless, the combined CO and ozone evaluation now provides a more consistent and comprehensive picture than in CLaMS-1.0 (see lines 336-344 and Appendix E in the revised version).

**Major Comment 4: Evaluation limited to climatologies; add case study**

*Reviewer:* The evaluation relies mainly on zonal and seasonal climatologies. While adequate for large-scale diagnostics, it limits assessment of the convective scheme under specific meteorological conditions. Including a specific case study (e.g., a well-observed convective event) would provide a more convincing evaluation.

   **Author response:** As many previously published CLaMS studies (see e.g.: Pommrich et al. (2014); Vogel et al. (2016); Tao et al. (2018); Konopka et al. (2022)) have relied on case studies for model validation, it was somewhat intentional in this study to take a different approach. The main drawback of focusing on a specific convective event in a single geographic location

is that it does not necessarily demonstrate the general robustness or improvement of the model. In contrast, our approach based on MLS and ACE-FTS climatologies allows for a more systematic, global, and multi-seasonal evaluation of convective transport across various regimes. Although this comes at the expense of resolving small-scale features typically observed in aircraft campaigns, it offers a broader validation framework. Since our paper is already quite long (35 pages), we prefer not to include an additional case study here, but we refer the reviewer to other recent CLaMS studies mentioned above where such case-specific comparisons were performed.

**Major Comment 5: Quantitative mass conservation assessment**

*Reviewer:* Section 2.4 raises the issue of mass conservation. However, the discussion remains qualitative. Could the authors provide a quantitative assessment of mass conservation errors, ideally as a function of region or meteorological regime? I suspect the largest errors may occur in regions with strong convection.

   **Author response:** We appreciate this important point. Mass conservation is a particularly vulnerable aspect of Lagrangian transport modeling, as it is not guaranteed by construction. We are currently working on a separate project aimed at quantifying this issue and improving mass conservation in CLaMS-3.0, particularly along the lines outlined in Appendix C. This includes implementing Voronoi-based weighting during the adaptive grid interpolation to ensure more rigorous mass consistency.

Since this work is still in progress and not yet ready for publication, we prefer to postpone a full quantitative assessment and any related figures to a forthcoming dedicated study. A promising test case is the recent Hunga Tonga eruption and the associated water vapor injection into the stratosphere. Because water vapor behaves approximately as a passive tracer and is well observed by MLS, it offers an ideal benchmark for assessing mass conservation in CLaMS. We expect to address this in detail in a follow-up paper.

**Technical comment: Figure 8 units**

*Reviewer:* Figure 8: There appears to be a unit error. The CO mixing ratio should be expressed in ppbv, not ppmv.

   **Author response:** Corrected.

**References**

Konopka, P., Tao, M., Ploeger, F., Diallo, M., and Riese, M. (2019). Tropospheric mixing and parametrization of unresolved convective updrafts as implemented in the chemical lagrangian model of the stratosphere (clams v2.0). *Geosci. Model Dev.*, 12:2441–2462.

Konopka, P., Tao, M., von Hobe, M., Hoffmann, L., Kloss, C., Ravegnani, F., Volk, C. M., Lauther, V., Zahn, A., Hoor, P., and Ploeger, F. (2022). Tropospheric transport and unresolved convection: numerical experiments with clams 2.0/messy. *Geoscientific Model Development*, 15(19):7471–7487.

Pommrich, R., Müller, R., Grooß, J.-U., Konopka, P., Ploeger, F., Vogel, B., Tao, M., Hoppe, C. M., Günther, G., Spelten, N., Hoffmann, L., Pumphrey, H.-C., Viciani, S., D'Amato, F., Volk, C. M., Hoor, P., Schlager, H., and Riese, M. (2014). Tropical troposphere to stratosphere transport of carbon monoxide and long-lived trace species in the chemical lagrangian model of the stratosphere (clams). *Geoscientific Model Development*, 7(6):2895–2916.

Tao, M., Pan, L. L., Konopka, P., Honomichl, S. B., Kinnison, D. E., and Apel, E. C. (2018). A lagrangian model diagnosis of stratospheric contributions to tropical midtropospheric air. *Journal of Geophysical Research: Atmospheres*, 123(17):9764–9785.

Vogel, B., Günther, G., Müller, R., Grooß, J.-U., Afchine, A., Bozem, H., Hoor, P., Krämer, M., Müller, S., Riese, M., Rolf, C., Spelten, N., Stiller, G. P., Ungermann, J., and Zahn, A. (2016). Long-range transport pathways of tropospheric source gases originating in asia into the northern lower stratosphere during the asian monsoon season 2012. *Atmospheric Chemistry and Physics*, 16(23):15301–15325.